# FREQUENCY-AWARE MASKED AUTOENCODERS FOR MULTIMODAL PRETRAINING ON BIOSIGNALS

## ABSTRACT

Leveraging multimodal information from biosignals is vital for building a comprehensive representation of people's physical and mental states. However, multimodal biosignals often exhibit substantial distributional shifts between pretraining and inference datasets, stemming from changes in task specification or variations in modality compositions. To achieve effective pretraining in the presence of potential distributional shifts, we propose a frequency-aware masked autoencoder (`bio`FAME) that learns to parameterize the representation of biosignals in the frequency space. `bio`FAME incorporates a frequency-aware transformer, which leverages a fixed-size Fourier-based operator for global token mixing, independent of the length and sampling rate of inputs. To maintain the frequency components within each input channel, we further employ a frequency-maintain pretraining strategy that performs masked autoencoding in the latent space. The resulting architecture effectively utilizes multimodal information during pretraining, and can be seamlessly adapted to diverse tasks and modalities at test time, regardless of input size and order. We evaluated our approach on a diverse set of transfer experiments on unimodal time series, achieving an average of ↑5.5% improvement in classification accuracy over the previous state-of-the-art. Furthermore, we demonstrated that our architecture is robust in modality mismatch scenarios, including unpredicted modality dropout or substitution, proving its practical utility in real-world applications. Code will be available soon.

## 1 INTRODUCTION

Physical and mental states of an individual are manifested by a variety of physiological responses or *biosignals*. For example, electroencephalography (EEG) can decode human emotions by monitoring their brain activities (Liu et al., 2010), electromyography (EMG) can detect facial expressions such as smiling by recording muscle contractions (Canento et al., 2011), and a combination of these modalities can help decode a person's affective states. The effective use of multimodal information can not only build better and more resilient representations of the human body and mental states (Bachmann et al., 2022; Smith & Gasser, 2005; De Sa & Ballard, 1998), but also help researchers understand how each biosignal contributes to each physiological state and how their information overlaps (Bird et al., 2020).

Recently, in language-vision domains, large-scale multimodal pretraining has demonstrated remarkable generalization and zero-shot capabilities (Huang et al., 2021; Bachmann et al., 2022; Radford et al., 2021), outperforming small-scale models that are trained on specific downstream tasks (Kirkpatrick et al., 2017; Radford et al., 2019). In light of these advancements, we investigate whether similar pretraining can be applied to the biosignal domain. However, performing multimodal pretraining on biosignals is particularly challenging due to the significant distributional shifts between the pretraining and downstream datasets. This challenge can be categorized in two ways: (i) For biosignals, there are substantial distributional **shifts within each modality**, wherein data varies across tasks, subjects, and even recording sessions within subjects due to slight changes in sensor placement and recording conditions (Cheng et al., 2020). Additionally, (ii) multimodal biosignals might encounter strong distributional **shifts across modalities**, meaning that the connection between different modalities can be altered. These crossmodal domain shifts can arise from unimodal shifts, as a change in a single modality can disrupt its relationship to a different modality. Moreover, multimodal biosignals often face *modality mismatch scenarios*, where modalities may be unavailable at

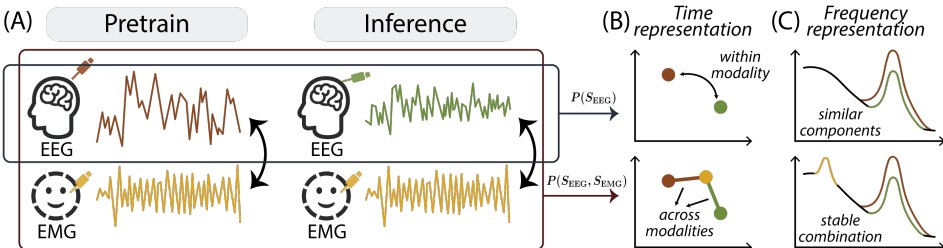

Figure 1: *Motivation of our approach.* **(A)** In multimodal biosignal systems, there exists substantial distributional shifts between the pretraining and inference datasets. **(B)** The distributional shifts often cause the shifts of representation in time-space, which can affect the model's generalization ability within modality and across modalities. **(C)** In the meantime, the representation in frequency-space typically would contain similar frequency components within modality, leading to more stable combinations in multimodal scenarios.

test time, and thus are removed or replaced with new modalities that provide relevant information to the detected physiological response (McKinzie et al., 2023). Addressing these distributional shifts is crucial to effectively leverage multimodal pretraining on biosignals.

In this work, we propose to incorporate frequency information in time series to mitigate distributional shifts and enable multimodal pretraining on biosignals. Frequency-domain analysis is advantageous for biosignals not only due to its invariance to common causes of distributional shifts such as temporal shifts and scaling, but also because the extracted frequency components are characteristic representations for physiological activities (see Figure 1). While previous works have shown the effectiveness of using frequency domain information to address generalization issues, they have either relied on encoders from both the time and frequency domains (Zhang et al., 2022b), or complicated sampling and combining modules (Zhou et al., 2022b) to utilize the frequency information. Here, we propose a simple, yet effective, multi-head frequency filter layer with fixed-size Fourier-based operator that directly parameterizes the representation of biosignals in the frequency space. The proposed layer can be easily incorporated into the transformer, giving a *frequency-aware (FA) encoder* that is both expressive and computationally efficient.

Furthermore, to extend the frequency awareness into a multimodal pretraining setting, we couple the FA encoder with a *frequency-maintain (FM) pretraining strategy*. To prevent the statistical consistency within the data from being disrupted by conventional masked autoencoding strategies (Ryali et al., 2023), our method performs masked autoencoding in the latent space to maintain the frequency awareness during reconstruction. Coupled with a channel-independent design (Nie et al., 2022; Liu et al., 2022b), our model presents a pure reconstruction-based multimodal pretraining architecture that can effectively combine and utilize information across modalities, with robustness towards distributional shifts within and across modalities.

To systematically evaluate our proposed approach, we first examine the transferability of our architecture on a publicly available one-to-many transfer learning benchmark (Zhang et al., 2022b). Our architecture achieves state-of-the-art performance, giving an average of ↑5.5% improvements in classification accuracy over the previous state-of-the-art, showing consistency across datasets of different input lengths, sampling rates, and diverse sources of modalities. Next, we demonstrate that our architecture is robust to a variety of modality mismatch scenarios commonly encountered in real-world cases, showing that our architecture can effectively integrate and leverage information across multiple modalities during pretraining.

We summarize our main contributions as follows:

- We propose `bioFAME`, a frequency-aware masked autoencoder for biosignals comprising: (i) a frequency-aware (FA) transformer encoder that can learn biosignals in a robust and computationally efficient way; (ii) a frequency-maintain (FM) pretraining strategy that retains the frequency awareness during reconstruction.
- By constructing a fixed-size Fourier-based operator in the architecture, `bioFAME` can be pretrained on multimodal biosignals and adapted to new modalities of varying lengths and frequency components, exhibiting resilience to distributional shifts even when the modalities differ between training and testing.

- `bioFAME` achieves consistently robust performance on a diverse set of transfer experiments, outperforming the previous state-of-the-art by an average improvement of ↑5.5%, demonstrating how utilizing multimodal information at the pretraining stage can benefit the generalization ability of the model.

## 2 BACKGROUND

**Multimodal Pretraining Methods**   Pretraining large-scale models that can effectively use multimodal information has gathered a lot of research attention due to its strong capability of generalization (Huang et al., 2021; Liang et al., 2022; Reed et al., 2022; Chai & Wang, 2022). Multimodal pretraining methods can be roughly categorized as (i) those that train separate encoders for each modality, as seen with contrastive methods that design novel objectives to align or fuse representations from different modalities (Li et al., 2021a; Radford et al., 2021; Jia et al., 2021), and (ii) those that design one unified architecture for many modalities, with completely shared encoders per-modality or a few layers shared for decoding (Reed et al., 2022; Akbari et al., 2021; Wang et al., 2022). The benefit of using one unified architecture is that we can build a joint representation space that connects different modalities, as well as share weights to reduce additional computational overhead (Bachmann et al., 2022; Lu et al., 2022). Inspired by the latter, our work aims to train a single unified architecture for multimodal biosignals with an effective frequency-awareness design.

**Pretraining on Biosignals and Time Series**   Biosignals are multivariate time series that capture various physiological processes within the human body (Giannakakis et al., 2019; Cheng et al., 2020). While biosignals are crucial for diverse applications such as human-computer interaction, acquiring an ample amount of labeled biosignals is a labor-intensive process that requires the involvement of domain experts (Ericsson et al., 2022). To alleviate the need for labeled data, researchers proposed various self-supervised methods to pretrain the model with large-scale unlabeled datasets. This includes (i) contrastive methods that build latent representation based on similarity across samples of different augmentation (Cheng et al., 2020; Kiyasseh et al., 2021; Zhang et al., 2022b), (ii) reconstruction-based methods that perform either feature reconstruction or data reconstruction (Kostas et al., 2021; Chien et al., 2022), or (iii) a hybrid of both (Dong et al., 2023). While previous works demonstrate that pretraining on large-scale data can benefit downstream task performance, however, most of the existing works only explored unimodal pretraining without investigating how to effectively utilize the multimodal information present at training time. Existing work even shows that pretraining on multimodal information could cause performance degradation due to the large variation across modalities (Zhang et al., 2022b). To the best of our knowledge, this is the first work that explores how to effectively perform multimodal pretraining on biosignals that gives robust performance towards distributional shifts within and across modalities.

## 3 MOTIVATION OF OUR APPROACH

Parameterizing representations in the frequency space is shown to be effective in many domains. Frequency-based approaches are particularly effective in solving partial differential equations and modeling long sequences (Li et al., 2020b; Gu et al., 2021; Li et al., 2022b; Zhou et al., 2022a), as it can effectively capture long-range dependencies. Frequency-aware approaches are also widely used in computer vision, as it can improve image fidelity and can effectively mix tokens when used in the transformer architecture (Rao et al., 2021; Guibas et al., 2021; Xie et al., 2022; Liu et al., 2022a; Li et al., 2022a). Akin to physiological signal processing, frequency-based approaches are employed to effectively extract discriminative patterns within sensory signals (Yao et al., 2019; Li et al., 2021b). The robustness of frequency-based operations can be partially attributed to the connection between Fourier transform and global circular convolution (Zhi et al., 2016; Li et al., 2020a).

Recently, many works suggest that the periodic oscillations and analogous patterns in the frequency space exhibit rich information for electrophysiological signals (Donoghue et al., 2020; Bird et al., 2020; Subha et al., 2010; Demanuele et al., 2007). Thus, several frequency-aware approaches are proposed to study biosignals. For example, Zhang et al. (2022b) used the consistency between time and frequency spaces to guide the learning on biosignals, demonstrating improved transferability and generalizability on downstream tasks. Other works perform cross-domain reconstruction across the time and spectral domains (Zhang et al., 2022a; Yang & Hong, 2022).

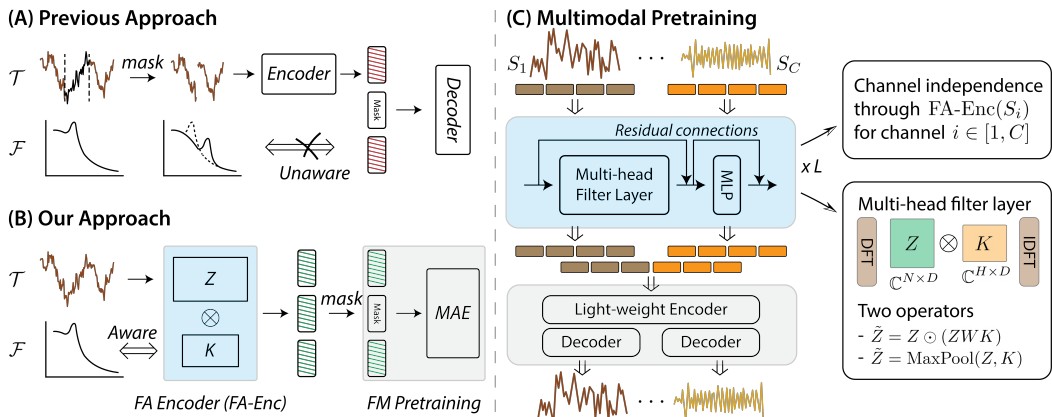

Figure 2: *Overview*. **(A)** Previous approaches perform masking in the time domain, which causes shifts in the frequency components. Also, the encoders are unaware of the frequency information in time series. **(B)** To address the issues, we propose `bioFAME`, which (i) builds frequency awareness by directly learning frequency filters in the representation space, and (ii) performs masked autoencoding in the latent space to maintain frequency information during pretraining. **(C)** We implement `bioFAME` in the multimodal pretraining scheme, where the frequency-aware encoder (FA-Enc($\cdot$)) processes signals in a channel-independent manner, and extracts representations with multi-head filter layer with fixed-size Fourier operators. The frequency-maintain pretraining strategy further performs masked autoencoding in the latent space with separate reconstruction to guide the effective mixing of multimodal information.

Contrary to prior studies, `bioFAME` emphasizes transferability and efficient adaptation to downstream tasks across many physiological modalities, by leveraging frequency-space information during pretraining on multimodal data to forge a universal representation of biosignals. We design novel mechanism and architecture to build a fully transferable and computation-efficient approach for frequency-aware representation extraction, setting `bioFAME` apart from conventional methods that are constrained by frequency-space encoders or decoding components tailored to specific input sizes (Wu et al., 2022). These conventional methods often struggle with modality transfer due to varying frequency components and introduce unnecessary computational burdens and overparameterization. Our approach, in contrast, ensures flexibility and efficiency, free from such limitations.

## 4 METHOD

**Preliminaries: Discrete Fourier Transform (DFT) for Token Mixing** DFT is widely used in traditional methods for processing biosignals and images (Pitas, 2000). For a time space representation $\boldsymbol{x} \in \mathbb{R}^N$ with $N$ elements $x_n$, $n \in [0, N-1]$, its corresponding frequency space representation $\boldsymbol{z} \in \mathbb{C}^N$ with elements $z_k$ is produced by DFT ($\mathcal{F}(\boldsymbol{x}) = \boldsymbol{z}$), which can be inversed through the Inverse Discrete Fourier Transform (IDFT) ($\mathcal{F}^{-1}(\boldsymbol{z}) = \boldsymbol{x}$) as below:

$$\text{DFT:} z_k = \sum_{n=0}^{N-1} x_n e^{-i(2\pi/N)kn}, \quad \text{IDFT:} x_n = \frac{1}{N} \sum_{k=0}^{N-1} z_k e^{i(2\pi/N)kn}, \quad (1)$$

where $i$ is the imaginary unit. The computational complexity of DFT can be reduced from quadratic to $\mathcal{O}(N \log N)$ when leveraging the fast Fourier transform (FFT) algorithm (Brigham, 1988).

Consider a sequence $X = [\boldsymbol{x}_1, ..., \boldsymbol{x}_N]^T \in \mathbb{R}^{N \times D}$ of $N$ tokens of $D$-dimensions, transformers aim to learn the interactions across tokens, typically through the self attention operation. Recently, mixing tokens with frequency-based operations through DFT and IDFT is shown to be a computationally efficient alternative (Rao et al., 2021; Guibas et al., 2021), as it considers global-wise information mixing. The token mixing process is theoretically grounded by the Fourier Neural Operators (Li et al., 2020b), which is often implemented in its discrete form (denote as $\mathcal{K}$) as such:

$$(\mathcal{K}(X))(\boldsymbol{x}_i) = \mathcal{F}^{-1}(R \cdot \mathcal{F}(X))(\boldsymbol{x}_i), \forall i \in [1, N] \quad (2)$$

Ideally, $R$ should be the Fourier transform of a periodic function which admits a Fourier series expansion. For the sake of simplicity, it is often implemented as learnable weights of shape $\mathbb{C}^{N \times D}$.

## 4.1 FREQUENCY-AWARE TRANSFORMER WITH MULTI-HEAD FREQUENCY FILTERS

In this work, we seek to understand two questions: (i) If parameterizing biosignals in the frequency space would provide better empirical performance, as frequency information is shown to be vital for many physiological activities; (ii) How to design a frequency-aware architecture that is transferrable and generalizable across different types of biosignals with varying input lengths and sampling rates. To address those two questions, we propose a multi-head frequency filter layer to build a frequency-aware transformer encoder $\text{FA-Enc}(\cdot)$.

**Multi-head Frequency Filter Layer**   We propose to manipulate the frequency representation with a multi-head frequency filters $K \in \mathbb{C}^{H \times D}$, where $H$ is the total number of heads. Given a sequence of tokens $X \in \mathbb{R}^{N \times D}$, we first perform DFT along the sequence dimension to obtain its representation in the frequency space as $Z \in \mathbb{C}^{N \times D}$. To obtain the manipulated features in frequency space $\tilde{Z} \in \mathbb{C}^{N \times D}$, we first compute queries $Q = ZW$, where $W \in \mathbb{R}^{D \times H}$ is a learnable matrix that is used to combine processed information across different filters. The resulting queries are used to re-weight the kernels to obtain $\tilde{Z}$ through the below operations:

$$\tilde{Z} = Z \odot (QK) = Z \odot (ZWK) \tag{3}$$

where $\odot$ is the Hadamard product. We show in Appendix C that the operation is equivalent to a weighted summation between each modulated frequency representation matrix, where the weights are self-generated through the queries. We note that our proposed operation, different from (Rao et al., 2021; Guibas et al., 2021), is applicable on time series with dramatic changes in input lengths and sampling rates, as we use a flexible fixed-sized multi-head filters $K$ that enables the transferability of the model. Intuitively, the querying process has similarity to hypernetworks (David et al., 2016), which generates weights based on data itself to fully exploit the structure of the data.

Having successfully incorporated a fix-sized multi-head filter $K$ into the frequency space, we further explored to build nonlinearity into the operation through an alternative maxpooling operation $\tilde{Z} = \text{MaxPool}(Z, K)$:

$$\tilde{Z}[i, j] = \max_k |Z[i, j]K[k, j]| \tag{4}$$

where the max-pooling is performed based on the absolute value of the complex features.

The resulting modulated frequency representation $\tilde{Z}$ is later recovered in time space through $\tilde{X} = \mathcal{F}^{-1}(\tilde{Z})$ with IDFT (see Figure 2(C)). We denote the whole process as $\text{Freq-L}(\cdot)$, which is computationally efficient, transferrable across different input lengths and sampling rates, and can be easily implemented in a few lines of code.

**Add** $\text{Freq-L}(\cdot)$ **into the Transformer**   The transformer architecture has revolutionized many domains, including natural language processing (Devlin et al., 2018), computer vision (Dosovitskiy et al., 2020), and recently time series processing (Nie et al., 2022). Following Nie et al. (2022), we first patchify the biosignals by dividing them into chunks, compute representations for each patch, and then feed the resulting patches into a transformer. Specifically, for a signal $\mathbf{s} \in \mathbb{R}^L$ where $L$ is the total length of the sequence, we divide them into sequences of $S = [\mathbf{s}_1, ... \mathbf{s}_N]$, where each patch $\mathbf{s}_i \in \mathbb{R}^P$ has a size of $P$. An initial MLP is used to compute representation $\mathbf{x}_i = \text{MLP}(\mathbf{s}_i) \in \mathbb{R}^D$, and the sequence is later stacked into $X_0 \in \mathbb{R}^{N \times D}$.

We replace the multi-head self-attention with our proposed multi-head frequency filter layer $\text{Freq-L}(\cdot)$ to mix the information across the sequence of tokens, which gives the FA transformer encoder layer as below:

$$X_{\ell+1} = X_\ell + \text{Freq-L}\left(X_\ell\right) + \text{FF}\left(X_\ell + \text{Freq-L}\left(X_\ell\right)\right), \ell = \{0, \ldots, L-1\} \tag{5}$$

where the representation is passed into the proposed $\text{Freq-L}(\cdot)$ layer and projection layers $\text{FF}(\cdot)$ with residual connections, as shown in Figure 2(C).

## 4.2 FREQUENCY-MAINTAIN PRETRAINING WITH LATENT MASKING AND CHANNEL INDEPENDENCE

**Masked Autoencoding in the Latent Space**   Masked autoencoder (MAE) is a self-supervised pretraining framework, which masks out input patches and predicts the missing patches using the

rest present patches. The architecture typically contains an transformer encoder that processes non-masked patches, follows by a decoder, usually a lightweight transformer, that reconstructs the original patches (He et al., 2022).

To preserve the frequency information while being able to perform pretraining based on the masked autoencoding strategy, we perform *masked autoencoding in the latent space*. Specifically, denote our frequency-aware transformer encoder as FA-Enc($\cdot$), full sequence of biosignals $S$ is learnt through FA-Enc($\cdot$) to obtain $X_L = [\boldsymbol{x}_1^L, \boldsymbol{x}_2^L, ..., \boldsymbol{x}_N^L]$. We sample a random set of patches based on a fixed masking ratio without replacement, and then process the resulting sequence with a lightweight transformer (second) encoder. We later pad the masked patches with mask tokens, and pass the resulting sequence into a lightweight transformer decoder to reconstruct the original signal, where the $i$-th reconstructed patch corresponds to $\boldsymbol{s}_i$. Denote the masked autoencoder as MAE($\cdot$), bioFAME aims to optimize the below objective:

$$\mathcal{L} = \frac{1}{\Omega} \sum_{i \in \Omega} l(\boldsymbol{s}_i, \text{MAE}(\text{FA-Enc}(S))[i]) \tag{6}$$

where $i$ is the token index, $\Omega$ is the set of masked tokens, and $l$ is an error term which is set as mean squared error (MSE) in this work. We show in Section 5 that the performance is robust if we remove MAE($\cdot$) and only keep FA-Enc($\cdot$) at test time. We note that this is the first work that finds using the masked autoencoding objective itself, without any contrastive terms, is effective on biosignals (Zhang et al., 2022b).

**Channel and Modality Independence**  Biosignals are multivariate time series that often face channel-wise and modality-wise mismatch at test time. To obtain robust transfer performance, we follow previous works to use *channel-independent design* before the second encoder to model multimodal biosignals (Liu et al., 2022b; Nie et al., 2022).

Given a multi-channel biosignal $[S_1, S_2, ..., S_C]$, where $C$ denotes the total amount of channels. We perform the channel independence learning such that each $S_\xi$ are passed into FA-Enc($\cdot$) and MAE($\cdot$) as below:

$$\mathcal{L} = \frac{1}{\Omega} \sum_{i \in \Omega} l(\boldsymbol{s}_i, \text{MAE}([\text{FA-Enc}(S_1), ..., \text{FA-Enc}(S_C)])[i]) \tag{7}$$

where $\Omega$ is the union of masked tokens for each channels, which is independently determined based on a fixed masking ratio for each channel. The parameter weights of the frequency-aware transformer encoder FA-Enc($\cdot$) are shared across channels, creating representations that are fed into the MAE($\cdot$), which combines information from different pretraining modalities. By combining the channel independence design into our multimodal masked autoencoding objective, our architecture can process input signals of any channel size and order, making it robust to multimodal distributional shifts when modalities are unavailable at test time.

## 5 EXPERIMENTS

### 5.1 TRANSFER EXPERIMENTS ON UNIMODAL TIME SERIES

**Datasets**  We first evaluate the model's generalization ability by transferring it on a diverse set of unimodal time series downstream tasks, following Zhang et al. (2022b). The transfer experiments include a set of four downstream tasks: Epilepsy (Andrzejak et al., 2001) (EEG measurement of disordered brain activity, sampling rate 174Hz with length 178); SleepEOG (Kemp et al., 2000) (EOG measurement of each sleep stage, sampling rate 100Hz with length 3000); ExpEMG (Goldberger et al., 2000) (EMG measurement of muscular disorders, sampling rate 4000Hz with length 1500); FD-B (Lessmeier et al., 2016) (Electromechanical measurement of motor disorder, sampling rate 64000Hz with length 5120). We performed data pre-processing following the same protocol and data split as in Zhang et al. (2022b), more details are in Appendix B.1. For model pretraining, we used the SleepEDF dataset (Kemp et al., 2000) as in (Eldele et al., 2021; Zhang et al., 2022b), where the single-channel EEG (channel Fpz-Cz) is commonly used for unimodal pretraining. In this work, we also used an additional EEG channel (Pz-Oz) and an additional modality (EOG) from SleepEDF to perform multimodal pretraining with the same train/test split as in Eldele et al. (2021).

*I. Generalization with modality or task association.*

| | Epilepsy (EEG) | | | | SleepEOG | | | |
|---|---|---|---|---|---|---|---|---|
| Models | Accuracy | Precision | Recall | F1 | Accuracy | Precision | Recall | F1 |
| TS-SD (Shi et al., 2021) | 80.18 | 76.47 | 89.52 | 77.67 | 48.90 | 28.59 | 25.43 | 23.68 |
| Mixing-up (Wickstrøm et al., 2022) | 80.21 | 40.11 | 50.00 | 44.51 | - | - | - | - |
| TS2vec (Yue et al., 2022) | 93.95 | 90.59 | 90.39 | 90.45 | 67.90 | 58.23 | 62.15 | 59.28 |
| CLOCS (Kiyasseh et al., 2021) | 95.07 | 93.01 | 91.27 | 92.06 | 66.86 | 56.67 | 58.99 | 57.34 |
| TS-TCC (Eldele et al., 2021) | 92.53 | 94.51 | 81.81 | 86.33 | 69.65 | 61.56 | 61.49 | 61.16 |
| TF-C (Zhang et al., 2022b) | 94.95 | **94.56** | 89.08 | 91.49 | 69.58 | 62.04 | 68.05 | 64.15 |
| PatchTST (Nie et al., 2022) | 95.01 | 91.66 | **92.96** | 92.27 | 68.00 | 61.20 | **68.28** | 63.26 |
| **bioFAME** (scratch) | 90.41 | 84.64 | 86.29 | 85.33 | 68.29 | 60.03 | 66.10 | 61.81 |
| **bioFAME** (unimodal) | 95.51 | 94.02 | 91.57 | **92.72** | 70.03 | 63.37 | 68.00 | 65.05 |
| **bioFAME** (multimodal) | **95.71** | 93.57 | **92.82** | **93.18** | **71.55** | **64.80** | **68.70** | **66.62** |
| Δ(uni, multi) | ↑0.20 | ↓0.45 | ↑1.25 | ↑0.46 | ↑1.52 | ↑1.43 | ↑0.70 | ↑1.57 |

*II. Generalization without explicit association.*

| | ExpEMG | | | | FD-B (Electromechanics) | | | |
|---|---|---|---|---|---|---|---|---|
| Models | Accuracy | Precision | Recall | F1 | Accuracy | Precision | Recall | F1 |
| TS-SD (Shi et al., 2021) | 46.06 | 15.45 | 33.33 | 21.11 | 55.66 | 57.10 | 60.54 | 57.03 |
| Mixing-up (Wickstrøm et al., 2022) | 30.24 | 10.99 | 25.83 | 15.41 | 67.89 | 71.46 | 76.13 | 72.73 |
| TS2vec (Yue et al., 2022) | 78.54 | 80.40 | 67.85 | 67.66 | 47.90 | 43.39 | 48.42 | 43.89 |
| CLOCS (Kiyasseh et al., 2021) | 69.85 | 53.06 | 53.54 | 51.39 | 49.27 | 48.24 | 58.73 | 47.46 |
| TS-TCC (Eldele et al., 2021) | 78.89 | 58.51 | 63.10 | 59.04 | 54.99 | 52.79 | 63.96 | 54.18 |
| TF-C (Zhang et al., 2022b) | 81.71 | 72.65 | 81.59 | 76.83 | 69.38 | 75.59 | 72.02 | 74.87 |
| PatchTST (Nie et al., 2022) | 92.68 | 90.87 | 94.51 | 92.07 | 67.03 | 71.96 | 75.57 | 70.09 |
| **bioFAME** (scratch) | 93.17 | 88.58 | 94.10 | 89.97 | 67.92 | 76.45 | 76.51 | 76.20 |
| **bioFAME** (unimodal) | **98.05** | **97.07** | 96.63 | 96.40 | **76.58** | **83.28** | **82.85** | **82.63** |
| **bioFAME** (multimodal) | **98.54** | 96.67 | **98.95** | **97.64** | **78.18** | **84.99** | **84.01** | **83.75** |
| Δ(uni, multi) | ↑0.49 | ↓0.40 | ↑2.32 | ↑1.24 | ↑1.60 | ↑1.71 | ↑1.16 | ↑1.12 |

Table 1: *Transfer experiments on unimodal time series*. All benchmark models are pretrained on the same single-lead EEG. All variants of our model is based on the same architecture, where bioFAME (scratch) is trained from scratch, bioFAME (unimodal) follows the same pretraining as baselines, and bioFAME (multimodal) is pretrained on the multimodal version of the data. Model standard deviation are in Appendix A.3.

**Experimental Details**  For bioFAME, we used a 4-layer encoder, 8-head filter with 64 dimensions. The model was trained using an Adam optimizer with $\beta_1 = 0.9$, $\beta_2 = 0.99$, and a learning rate of 0.001. We performed a grid search based on the validation set to select the model hyperparameters (see Appendix B.4). Following prior works, we performed full model fine-tuning on all tasks (see details in Appendix B.2). In contrast to state-of-the-art contrastive architectures (Eldele et al., 2021; Zhang et al., 2022b), we did not apply data augmentation in our architecture as we found there was minimal impact on performance. We repeated experiments with five random seeds for major results, and three random seeds for ablation experiments (see model variation in Appendix A.3). To benchmark our method, we selected an extensive set of existing state-of-the-art models, including temporal-spatial methods (Shi et al., 2021; Yue et al., 2022), contrastive methods (Kiyasseh et al., 2021; Eldele et al., 2021), transformers and frequency-aware approaches (Nie et al., 2022; Zhang et al., 2022b). All benchmark models were pretrained on unimodal EEG under the same data split, providing a conclusive list of models for fair comparison.

**Pretraining on Unimodality**  Following previous works Zhang et al. (2022b), we first performed pretraining on a single-channel EEG from the SleepEDF dataset, and then fine-tuning on a small amount of data from the downstream tasks. The performance of our proposed architecture is shown in Table 1. We show that with the same unimodal pretraining setup on single-channel EEG, our model consistently outperforms state-of-the-art benchmarks in most experiments, giving ↑4.2% improvments in accuracy. These results demonstrate that bioFAME is effective in terms of transfer on different tasks, with robustness to domain shifts across tasks, subjects, sampling rate, and sensors. Surprisingly, our architecture, without any pretraining (scratch), also provides robust performance on many datasets, different from previously reported results (Zhang et al., 2022b). This further demonstrates the robustness of our proposed architecture.

| FA | FM | Acc. |
|----|----|------|
| ✗ | ✗ | 80.68 |
| ✓ | ✗ | 84.09 |
| ✗ | ✓ | 83.53 |
| ✓ | ✓ | **85.04** |

Table 2: Average accuracy without FA/FM modules.

| Enc-2 | Modality | Acc. |
|-------|----------|------|
| ✗ | Uni | 85.04 |
|   | Multi | 83.92 |
| ✓ | Uni | 85.05 |
|   | Multi | **85.99** |

Table 3: The effect of keeping the 2nd encoder for multimodal pretraining.

| | | Masking ratio | | |
|---|---|------|------|------|
| | | 0.3 | 0.5 | 0.7 |
| *Patch* | 10 | 83.86 | 84.05 | 82.70 |
| | 20 | 84.11 | **85.04** | 83.86 |
| | 50 | 80.88 | 80.84 | 80.64 |

Table 4: The effect of different masking ratios and patch sizes.

**Extending Pretraining to Multimodality**   While the Fpz-Cz EEG channel is shown to be the most informative channel for the pretraining task and typically provides robust prediction performance on its own (Supratak et al., 2017), in this work, we explore whether using additional multimodal information from the same task can further boost the pretraining performance. As shown in Table 1, for bioFAME, including multimodal information during pretraining provides better results than unimodal pretraining in general, consistently outperforming unimodal pretraining. Training on multimodal data also improves the model's stability by giving a lower standard deviation, as shown in Appendix B.4. Note that in previous work (Zhang et al., 2022b), including multimodal information hurt performance rather than helped. This suggests that bioFAME can effectively utilize and combine information across modalities, resulting in better performance on downstream tasks. We hypothesize that pretraining on multiple modalities exposes the model to a more diverse range of frequency components, improving the model's few-shot generalization.

**Ablations Experiments on Transferability**   We performed a set of ablation experiments to understand what makes bioFAME robust under the transfer experiments setting (more in Appendix A.1). In Table 2, we first studied the effect of the frequency-aware (FA) and frequency-maintain (FM) modules by either replacing the FA module with a self-attention transformer; or by replacing the FM module with a normal masking procedure. We found both approaches, when applied independently, improve the performance of a baseline variant by a significant margin ($\approx 3\%$). Combining both modules gives the best performance, further boosting the effect of each individual component ($\approx 5\%$). We also tested whether it is possible to discard the second encoder at test time, which would indicate whether or not the FA encoder plays a major role in learning. Interestingly, we show that discarding the second encoder at test time gives almost identical performance in the unimodal setting. However, when multimodal information is used for pretraining, discarding the second encoder would give a performance that is lower than the unimodal result, while keeping the second encoder increases the unimodal performance by $\approx 1\%$ instead (see Table 3). We hypothesize that it is beneficial to retain the second encoder at test time under the multimodal setting because it is responsible for merging the information present across the multimodal data. Finally, in Table 4, we investigate how different patch sizes and masking ratios affect the performance of our model. We show that bioFAME gives stable performance when the patch size is relatively small, giving robust performance under a range of masking ratios.

## 5.2   MULTI-MODAL EVALUATIONS AND VISUALIZATIONS

**Datasets and Experimental Details**   After verifying the model's generalization ability on transfer tasks, we investigated how well the model performs when applied to real-world cases in which multimodal information is available at test time. To understand this, we systematically studied different combinations of the EEG Fpz-Cz, EEG Pz-Oz, EOG, EMG, and the respiration channels of the SleepEDF dataset (Kemp et al., 2000), which are simultaneously recorded. We followed the same train/val/test split as in Eldele et al. (2021) while attaching the multimodal information instead of using only the unimodal information. We utilized the same model setup as in Section 5.1, aside from that we follow Section 4.2 to expand the training and testing under multimodal designs with weight sharing and channel independence. We also implemented two variants of multimodal latent expansion methods as in Appendix C.

**Robustness for Modality Mismatch Scenarios**   We consider two modality mismatch scenarios as shown in Figure 3(A): (i) Modality substitution, where one modality is replaced by another modality; and (ii) Modality dropout, where only a subset of modalities is present at test time. We show the model's performance with modality substitution in Figure 3(B), where the model is pretrained

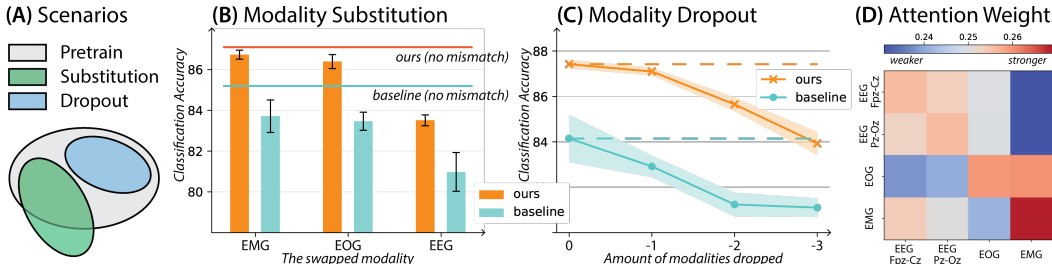

Figure 3: *Multimodal evaluation results*. **(A)** Two modality mismatch scenarios are considered: Modality substitution and modality dropout. **(B)** When a modality is swapped with another available one, or **(C)** when modalities are dropped out at test time, our model gives lower performance degradation when comparing to a robust baseline. **(D)** By visualizing the attention weights across modalities, we can understand how modalities are associated with each other.

with { EEG Fpz-Cz; EOG; EMG }. Each of the pretraining modality is replaced with another channel to examine the performance degradation (more details in Appendix B.3). Our model gives better performance than the robust baseline PatchTST (Nie et al., 2022), exhibiting less performance degradation. In terms of modality dropout, we pretrained the model with { EEG Fpz-Cz; EEG Pz-Oz; EOG; EMG }, and we dropped an increasing amount of modalities till there is only one modality left (see Figure 3(C)). We see that `bioFAME` is more resistant to unexpected modalities dropout in comparison to the baseline. Unlike many other baselines that contain spatial layers, `bioFAME` can be applied at test time even when there are unexpected amount of channels while exhibiting resilience towards modality mismatch scenarios. This study further demonstrated that `bioFAME` presents a robust model when used in real-world scenarios.

**Visualizing the Connections Across Modalities**  To understand how the information across different channels affects each other, we visualized the averaged attention matrix to examine the relationship across modalities. As shown in Figure 3(D), for each channel (row), the intensity of its attention or connection to the other channels can be visualized by the color (red means stronger connections). Interestingly, we notice that while each channel would rely on its own information the most, they tend to focus on the stronger modalities, which is the EEG Fpz-Cz channel in our case. Moreover, interesting asymmetry is observed for EOG-EMG, as EOG correlates more to the EMG while the opposite does not hold. We hypothesize that this is because facial movement would produce moving artifacts for EOG on the temple, while the opposite connection does not hold. This observation demonstrates that `bioFAME` can be used by researchers to further understand the information overlap across modalities (Bird et al., 2020).

## 6 CONCLUSION

In this work, we proposed a frequency-aware masked autoencoder that performs pretraining on multimodal biosignals. Our proposed method leverages a frequency-aware encoder with fixed-size Fourier-based operator to extract representation on biosignals, and uses a frequency-maintain pretraining module to perform pretraining. We performed extensive empirical experiments to show that (i) our model achieves state-of-the-art performance on a set of transfer experiments, where the models, both pretrained on unimodality and multimodality, can be adapted to effectively classify time series with varying input lengths, sensors, and sampling rates; and (2) our model demonstrates resilience to within-modal and across-modal distributional shifts, shows robust performance when applied in modality mismatch scenarios that are common in real-world applications.

While our model provides a good balance between utilizing frequency-information and operating on time domain, we note that, just like other frequency-aware architectures (Li et al., 2020b), it remains underexplored how to interpret the specific band and type of frequency information that is taking effect in each downstream task. Exploring how the learned frequency filters can be structured and interpreted will be an exciting line of future research. Also, in our current formulation, we only consider low-density biosignal recording systems due to the lack of publicly available high-dimensional multimodal biosignal datasets. Given the constraints, our architecture relies on the channel-independent design, which is known to suffer from capacity and robustness trade-off (Han et al., 2023). Extending and scaling our approach to high-dimensional sensor inputs is another exciting line of future research for modeling comprehensive human states.

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
