APPENDIX

# A    ADDITIONAL RESULTS

## A.1    PARAMETER EFFICIENCY AND ADDITIONAL ABLATIONS

**Parameter efficiency**    To understand the parameter efficiency and the throughput of our approach, we compute the parameters and FLOPs between baselines and our approach in Table 5.

|        | TS2vec | TFC   | TS-TCC | PatchTST | **Ours** |
|--------|--------|-------|--------|----------|----------|
| Params | 632K   | 1.18M | 140K   | 612K     | 243K     |
| FLOPs  | 0.69B  | 1.38B | 1.95B  | 35.0B    | 9.42B    |

Table 5: Comparison of parameters and FLOPs between baselines and our approach. The FLOPs are computed over a batch of SleepEDF data with batch size 64.

We can see that, `bioFAME` is very parameter-efficient due to its fix-size frequency filter design. With the same depth (4), heads (8), and dimensionality (64), bioFAME contains only $\approx 40\%$ parameters of the transformer baseline PatchTST. The parameter size of bioFAME also stands competitive with many CNN-based architectures. The FLOPs of bioFAME are significantly lower than the transformer baseline PatchTST ($<30\%$); yet greater than CNN-based architectures.

**Additional ablations**    To understand the models' sensitivity towards different hyperparameters and understand if `bioFAME` can provide better performance with increased complexity, we conducted additional ablation experiments in Table 6 and Table 7.

| dim    | 32    | 64    | 128   | 256   |
|--------|-------|-------|-------|-------|
| ExpEMG | 91.1  | 98.05 | 96.48 | 97.78 |
| FD-B   | 76.74 | 76.58 | 78.14 | 80.87 |
| Avg.   | 83.92 | 87.32 | 87.31 | 89.33 |

Table 6: Performance of our approach with different latent dimensionality.

| depth  | 3     | 4     | 5     | 6     |
|--------|-------|-------|-------|-------|
| ExpEMG | 77.54 | 76.58 | 76.79 | 78.99 |
| FD-B   | 97.78 | 98.05 | 95.55 | 92.59 |
| Avg.   | 87.66 | 87.32 | 86.17 | 85.79 |

Table 7: Performance of our approach with different encoder depth.

We observed that increasing the latent dimensionality could further improve the performance of our approach; while increasing the network depth gives no performance gains.

## A.2    DATA EFFICIENCY AND OPERATOR SELECTION

**Data efficiency**    To understand the behavior of `bioFAME` within the context of limited data availability, we conducted experiments aimed at gauging the architecture's efficacy when exposed to a reduced amount of labeled data during the finetuning phase. We show the performance of `bioFAME` in Figure 4(A), both with and without pretraining, where the performance of `bioFAME` is plotted when the amount of labeled data for downstream training varies from $5\%$ to $100\%$. Notably, in contrast to previous work (Eldele et al., 2021), wherein architecture performance substantially deteriorated with decreased labeled data, `bioFAME` achieves stable results with relatively low decay of performance even without pretraining. Furthermore, the pretrained version of `bioFAME` gives consistently robust performance across the spectrum of labeled data proportions. We hypothesize that modeling biosignals using the Fourier function group with frequency operators improves the data efficiency of models.

**Ablations on the two operators**  To validate the effectiveness of the Maxpool operator and the Query operator as described in Section 4.1, we examine the model's performance by varying the number of filters. We find that the Maxpool operator gives more stable results, while the Query operator seems to scale better to larger amount of filters.

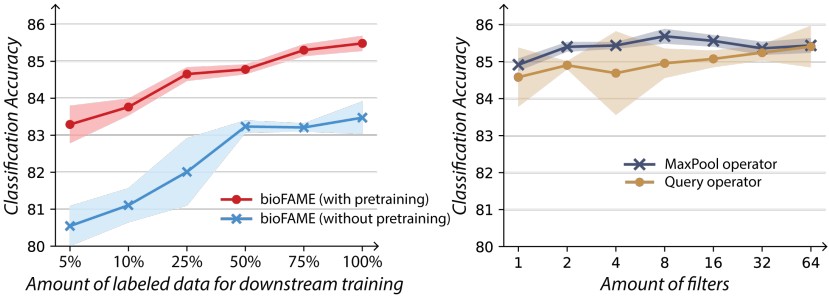

Figure 4: **(A)** We examine the performance of `bioFAME` under low-data regime with and without pretraining. **(B)** We examine how the MaxPool operator and Query operator would perform with different amounts of filters.

## A.3    MODEL VARIATION

For the transfer experiments result as shown in Table 1, we provide the standard variation across five different random seeds in Table 8. Note that the entire training process, both the pretraining and the finetuning stages, is repeated to obtain the standard variation for fair evaluation. We notice that multimodal pretraining typically gives a lower standard deviation than that of unimodal pretraining, demonstrating that multimodal pretraining might help with the stability of the model, as it is exposed to a variety of frequency components.

| Models | Epilepsy (EEG) | | | | SleepEOG | | | |
|---|---|---|---|---|---|---|---|---|
| | Accuracy | Precision | Recall | F1 | Accuracy | Precision | Recall | F1 |
| **bioFAME** (scratch) | 1.17 | 2.42 | 0.72 | 1.26 | 0.77 | 0.67 | 0.50 | 0.76 |
| **bioFAME** (unimodal) | 0.35 | 0.37 | 1.17 | 0.65 | 1.39 | 1.23 | 0.91 | 0.61 |
| **bioFAME** (multimodal) | 0.17 | 0.51 | 0.21 | 0.24 | 0.90 | 0.79 | 0.89 | 0.88 |
| $\Delta$(uni, multi) | ↓0.18 | ↑0.14 | ↓0.96 | ↓0.41 | ↓0.49 | ↓0.44 | ↓0.02 | ↑0.27 |

| Models | ExpEMG | | | | FD-B (Electromechanics) | | | |
|---|---|---|---|---|---|---|---|---|
| | Accuracy | Precision | Recall | F1 | Accuracy | Precision | Recall | F1 |
| **bioFAME** (scratch) | 2.67 | 3.13 | 2.25 | 3.15 | 1.63 | 1.33 | 1.20 | 1.09 |
| **bioFAME** (unimodal) | 2.04 | 2.80 | 5.64 | 4.15 | 2.74 | 1.75 | 2.01 | 2.14 |
| **bioFAME** (multimodal) | 1.34 | 3.04 | 0.96 | 2.15 | 1.94 | 1.53 | 1.44 | 1.66 |
| $\Delta$(uni, multi) | ↓0.70 | ↑0.24 | ↓4.68 | ↓2.00 | ↓0.80 | ↓0.22 | ↓0.57 | ↓0.48 |

Table 8: The standard deviation of `bioFAME` for each transfer experiment.

While we believe that our diverse experiments across many datasets demonstrate the robustness of our approach under randomness, we believe that another important source of randomness comes from the data split, which is fixed in this work.

## A.4    ABLATION RESULTS BREAKDOWN

In Table 9, we report the breakdown details for the average accuracy presented in Table 2 and Table 3. Our model provides robust performance across different downstream tasks consistently.

## B    EXPERIMENTAL DETAILS

## B.1    DATASETS DETAILS

We provide additional details about the datasets we used as follows.

|          | Ablations     | Epilepsy | SleepEOG | ExpEMG | FD-B  |
|----------|---------------|----------|----------|--------|-------|
|          | FA✗  FM✗      | 95.01    | 68.00    | 92.68  | 67.03 |
| Table 2  | FA✓  FM✗      | 95.03    | 69.73    | 98.37  | 73.23 |
|          | FA✗  FM✓      | 94.81    | 68.41    | 95.94  | 74.97 |
|          | FA✓  FM✓      | 95.51    | 70.03    | 98.05  | 76.58 |
|          | Uni, Enc-2✗   | 95.91    | 70.17    | 95.94  | 78.16 |
| Table 3  | Multi, Enc-2✗ | 95.26    | 71.04    | 96.10  | 73.28 |
|          | Uni, Enc-2✓   | 95.51    | 70.03    | 98.05  | 76.58 |
|          | Multi, Enc-2✓ | 95.71    | 71.55    | 98.54  | 78.18 |

Table 9: Breakdown of model performance on different downstream tasks.

**SleepEDF**   The entire SleepEDF dataset contains 197 recordings of whole-night sleep, where the dataset contains 2-lead EEG, EOG, chin EMG, respiration rates, body temperature, and event markers. We selected a subset of the dataset from the Cassette Study following Eldele et al. (2021), where the dataset is used to study the age effects on sleep in healthy Caucasians. We further followed the same train/validate/test split, and removed data with incomplete modalities. The recordings are segmented into 30 seconds of sleep for training, where each sample is associated with one of the five sleeping patterns/stages: Wake (W), Non-rapid eye movement (N1, N2, N3), and Rapid Eye Movement (REM).

**Epilepsy**   The Epilepsy dataset contains single-lead EEG measurements from 500 subjects, where the brain activities are recorded for subjects with seizure. The classification task is based on if the subject is having a seizure episode during the recording session.

**SleepEOG**   The SleepEOG dataset is a subset of the SleepEDF dataset under the Telemetry Study, where subjects are reported to have mild difficulty falling asleep, and thus intake either temazepam or placebo before sleep. The EOG channel is used for classification.

**ExpEMG**   The ExpEMG dataset consists of single-channel EMG recordings from the tibialis anterior muscle of three healthy volunteers, where they (1) do not have history of neuromuscular disease; (2) suffer from chronic low back pain and neuropathy; and (3) suffer from myopathy due to longstanding history of polymyositis. The classification task aims to classify different conditions (subjects).

**FD-B**   The FD-B dataset is an electromechanical dataset, where the motor currents and vibration signals of healthy or damaged motors are recorded. The classification task aims to detect different faulty conditions of the motors based on their behavior. We found that the motor movement follows a similar frequency assumption as biosignals (Hooge et al., 1981), and thus used this electromechanical dataset to provide additional validation of the transfer performance of our model.

| Datasets | Train | Validate | Test  | Sampling rate | Length |
|----------|-------|----------|-------|---------------|--------|
| Epilepsy | 60    | 20       | 11420 | 174           | 178    |
| SleepEOG | 1000  | 1000     | 37244 | 100           | 3000   |
| ExpEMG   | 122   | 41       | 41    | 4000          | 1500   |
| FD-B     | 60    | 21       | 13559 | 64000         | 5120   |

Table 10: Dataset split details for different downstream tasks.

We performed the transfer experiments based on the same settings as in Zhang et al. (2022b), where we used the train/validate/test spilt as shown in Table 10 for downstream fine-tuning to demonstrate the few-shot generalization ability of the model across different signals.

### B.2   MODEL TRAINING AND TRANSFER EXPERIMENTS DETAILS

For all experiments, we pretrain `bioFAME` for 200 epochs on the SleepEDF dataset using a batch size of 128 to obtain the weights of the model. During fine-tuning, we remove the lightweight second

encoder that mixes information across modalities, and use the average token of the frequency-aware transformer encoder to perform the prediction for downsteam tasks. We fine-tune `bioFAME` for 80 epochs with a batch size of 64, using an Adam optimizer with a learning rate of 0.001 on all datasets to obtain the final results. We perform all transfer experiments under the same training setup for all downstream tasks without additional adjustment for each dataset. Note that we perform full-scale model finetuning instead of linear probing when performing the transfer experiments, because the former approach is shown to be more effective for transformers in previous works (He et al., 2022).

## B.3 MULTIMODAL SETUP DETAILS

The multimodal experiments are designed to tackle the challenge presented by modality mismatch scenarios, where discrepancies in biosignal recording setups between training and testing phases lead to distributional shifts. Due to the scarcity of comprehensive multimodal datasets encompassing simultaneous recording of diverse modalities of biosignals, we exclusively used the SleepEDF dataset due to its modality coverage.

We first empirically assessed the representation quality of each individual channel. Similar to the findings in Supratak et al. (2017), we found that the representation capacity of different channels follows EEG Fpz-Cz > EEG Pz-Oz > EOG > EMG > resp. Building upon these insights, we performed the modality substitution and modality dropout experiments following the below pretraining and finetuning setup.

| Training modalities | Testing modalities |
|---|---|
| EEG Fpz-Cz; EOG; EMG | EEG Fpz-Cz; EOG; resp 
 EEG Fpz-Cz; EEG Pz-Oz; EMG 
 EEG Pz-Oz; EOG; EMG |

Table 11: Modality setup for modality substitution experiments.

| Training modalities | Testing modalities |
|---|---|
| EEG Fpz-Cz; EEG Pz-Oz; EOG; EMG | EEG Fpz-Cz; EEG Pz-Oz; EOG 
 EEG Fpz-Cz; EEG Pz-Oz 
 EEG Fpz-Cz |

Table 12: Modality setup for modality dropout experiments.

## B.4 HYPERPARAMETER SEARCHING DETAILS

For transfer experiments, we performed hyperparameter searching based on results on the Epilepsy dataset, and used the same parameter setting across all transfer experiments. Specifically, we performed a grid search of learning rate of [0.0001, 0.001, 0.01], transformer depth of [2, 3, 4, 5, 6], latent dimensionality of [16, 32, 64, 128], dropout rate of [0.2, 0.3, 0.4], operator type, and filter amount correspondingly. We followed the convention for transformers and selected the MLP dimension of 128 and head dimension of 16 for `bioFAME` and the baseline transformer. We selected the optimal patch size and masking ratio based on results in Table 4. We did not search for the optimal batch size, or investigate the effect of using different activation functions or normalization techniques. For multimodal experiments, we evaluate the model's performance on the pretraining dataset, and performed the evaluation on the finetuning modalities using the best model used in pretraining. For the multimodal experiments, we performed a smaller scale grid search for the latent dimensionality and transformer depth.

## C METHODOLOGY DETAILS

### C.1 ADDITIONAL EXPLANATION OF MOTIVATION

Biosignals are often analyzed in their frequency space, where they are either studied through predefined frequency regions or through aperiodic components which typically form a 1/f-like distribution (Donoghue et al., 2020). The significance of frequency information is well-documented due to its intricate interrelation with various facets of learning, aging, as well as diseases such as ADHD or

seizures. Correspondingly, modeling approaches that rely on the manual extraction and preprocessing of spectrogram features have demonstrated robust empirical performance (Supratak et al., 2017). Building upon these insights, we hypothesize that modeling biosignals employing function groups within the frequency domain could benefit the learning process by enhancing model adaptability and data efficiency. We note that this hypothesis might be violated if the frequency components carry limited information in other formats of time series datasets.

## C.2 INTUITION FOR THE MULTI-HEAD FREQUENCY FILTER LAYER

We provide additional intuition for the design of our multi-head frequency filter layer by breaking down the computation for each individual filter. For each $k$-th filter $K[k]$ inside $K \in \mathbb{C}^{H \times D}$, given latent representation $Z = [\boldsymbol{z}_1, \boldsymbol{z}_2, ..., \boldsymbol{z}_N]^T \in \mathbb{C}^{N \times D}$, we compute $Z^{(k)} = [\boldsymbol{z}_1 \odot K[k], \boldsymbol{z}_2 \odot K[k], ..., \boldsymbol{z}_N \odot K[k]]^T$, where $\odot$ represents the Hadamard product between each representation and the learnable filter weights. In order to learn the combination between different filters, we define weights $\boldsymbol{w}$ that compute $\tilde{Z} = \sum_{k=1}^H w_k Z^{(k)}$.

To increase the expressiveness of the filtering operation, instead of learning a linear combination of different filters, we borrow intuition from the computation of self-attention to compute the queries for the kernel weights $\boldsymbol{w}$ through $\boldsymbol{w} = \boldsymbol{z}W$, where $W \in \mathbb{C}^{D \times H}$. Thus, we have:

$$
\begin{aligned}
\tilde{Z}[i,j] &= \sum_{k=1}^H (\sum_{j=1}^D Z[i,j]W[j,k])Z[i,j]K[k,j] \\
&= Z[i,j] \sum_{k=1}^H (\sum_{j=1}^D Z[i,j]W[j,k])K[k,j]
\end{aligned}
\tag{8}
$$

which gives $\tilde{Z} = Z \odot (ZWK)$. In our implementation, we use the real values of latents to learn the weights of the combiner though $\boldsymbol{w} = \boldsymbol{z}_{\text{real}}W$. Similarly, based on the same intuition of combining filtered matrices, we have the max pooling operation.

## C.3 MODEL VARIANTS FOR COMBINING MULTIMODAL REPRESENTATIONS

In transfer experiments, we use the average of tokens to extract the final representations for the downstream classification. However, when having multimodal information, fixing the dimensionality of the latent representation when many modalities are present might narrow down the information from each modality, which might cause information loss. Thus, in multimodal experiments, we first average the representations from each individual modality, and then concat the representations across modalities before performing the downstream classification.