# OpenReview forum: "Frequency-Aware Masked Autoencoders for Multimodal Pretraining on Biosignals"
_ICLR.cc/2024/Conference — Submitted to ICLR 2024_

### Official Review · Reviewer_hfvK · 2023-10-25

**Soundness:** 2 fair
**Presentation:** 3 good
**Contribution:** 3 good
**Rating:** 6
**Confidence:** 4

**Summary:**

This paper focuses on the problem of learning a reliable pretraining approach against domain shifts in multimodal biosignal processing. Proposed method harnesses frequency-domain features in multimodal representation alignment, using a transformer-based masked autoencoder. Simulations are conducted on various multimodal biosignal datasets (e.g, EEG, EOG, EMG, ...), and the approach is demonstrated to be beneficial in several experiments.

**Strengths:**

- The approach is novel and effective in its application to multimodal representations and alignment in the frequency domain.
- The paper is written well with a good storyline and overview illustrations that give the main intuition clearly.

**Weaknesses:**

- Reproducibility in biosignal processing studies is often a concern. Therefore authors should consider strengthening their randomized experiment setup, and open sourcing their code and simulations reported in this submission.

**Questions:**

1) Regarding the downstream tasks: What is the true impact of the used dataset split folds? Did the authors investigate this in simulations where the randomness is controlled (they mention a number of seed repetitions here), or is the appendix Table 7 a one-time setup where test performances are only presented for? What is the performance deviation in such repetitions? This needs to be clarified a bit further to support the strength/reliability of the results.

2) How influential is the architecture capacity on the results? The design of the network backbone seems like an arbitrary choice, as it is not really justified much or studied in the ablations. In comparison to the other approaches, what would be the size of the network (or parameters to be optimized) in the proposed model?

3) What is the comparative computational overhead of bioFAME at training and test time, as opposed to previous alternatives (e.g., TS-TCC, PatchTST..)?

---

> ### Author Response · Authors · 2023-11-17
> **Response to Reviewer hfvK**
>
> Dear reviewer hfvK, we appreciate your time and insightful feedback. Especially, thank you for evaluating our work as “novel and effective”, and “well written”. We greatly appreciate your positive comments!
>
> ---
> ### **Reproducibility and randomness control**
>
> We totally agree that reproducibility is often a concern in the physiological signal field, and plan to publicize all related code soon with the exact settings. Thank you for your suggestion about the randomness control. While we have provided the model performance different across different random seeds under the same dataset splits in Appendix Table 5,  we agree that the data splits are another important source of randomness. However, because we performed many experiments across a diverse range of datasets, we believe the diversity in experiments could compensate for the data splits; and thus did not conduct additional experiments due to time constraints. We will make sure to note this in the limitation of the work.
>
> ---
> ### **Architecture capacity analysis and additional ablations**
>
> Thank you for your question regarding the architecture capacity and additional ablations!
>
> To understand the size of the model, we compare the total number of parameters (Params) and FLOPs of our approach and baselines.
>
> | Methods | TS2vec  | TFC | TS-TCC | PatchTST | **bioFAME** (Ours) |
> | -------- | ------- | -------- | ------- | ------- | ------- |
> | Params | 632K | 1.18M | 140K | 612K | 243K |
> | FLOPs | 0.69B | 1.38B | 1.95B | 35.0B | 9.42B |
>
> We can see that, bioFAME is very parameter-efficient due to its fix-size frequency filter design. With the same depth (4), heads (8), and dimensionality (64), bioFAME contains only ~40% parameters of the transformer baseline PatchTST. The parameter size of bioFAME also stands competitive than many CNN-based architectures, with only TS-TCC smaller than it.
>
> To gain a better understanding of the throughput of the model, we used the fvcore package to compute the FLOPs of the models over a batch of SleepEDF data (batch size = 64). We found that the FLOPs of bioFAME is significantly lower than the transformer baseline PatchTST (<30%); yet greater than CNN-based architectures.
>
> To understand the sensitivity of our architecture to hyperparameters, we also conducted additional ablation experiments to examine how the performance would change with respect to the changes in numbers of layers and the latent dimensionality. We observed that increasing the latent dimensionality could further improve the performance; while increasing the network depth gives little performance gains.
>
> | Latent dim | 32 | 64 | 128  | 256 |
> | -------- | ------- | -------- | ------- | ------- |
> | EMG | 91.1 | 98.05 | 96.48  | 97.78  |
> | FDB | 76.74  | 76.58 | 78.14  | 80.87  |
> | Avg. | 83.92 | 87.32 | 87.31 | 89.33 |
>
> | Network Depth | 3 | 4 | 5  | 6 |
> | -------- | ------- | -------- | ------- | ------- |
> | EMG | 77.54 | 76.58 | 76.79 | 78.99 |
> | FDB | 97.78 | 98.05 | 95.55 | 92.59 |
> | Avg. | 87.66 | 87.32 | 86.17 | 85.79 |

---

> ### Author Response · Authors · 2023-11-21
> **Follow-up to the Rebuttal**
>
> Dear reviewer,
>
> Thank you once again for your insightful feedback. We have carefully revised our manuscript in line with your suggestions and believe we have comprehensively addressed your concerns. Could you kindly confirm whether the revisions align with your expectations? We are available to engage in further discussion or answer any additional questions you may have.
>
> Thanks!

---

> > ### Comment · Reviewer_hfvK · 2023-11-23
> > **post-rebuttal response**
> >
> > Thanks to the authors for their responses.
> >
> > In general I'm happy with the way the manuscript is written, its novelty and scope. My only concern is on the significance of the results, which is still open. The fact that all experimental results are currently bound to certain train-test splits (App. Table 10) is slightly weighing in negatively. I think it should be somewhat simpler to just shuffle these arbitrary train/test splits of one dataset to run a few of the supplementary experiments in order to discard any suspicion on model's capabilities. The fact that there is more than one task/dataset (with again pre-defined arbitrary splits) does not really prove this generalization in the same sense. Therefore I am currently around the borderline, but in general at the positive side. I will keep my score as it is.

---

### Official Review · Reviewer_BHTD · 2023-10-31

**Soundness:** 2 fair
**Presentation:** 3 good
**Contribution:** 2 fair
**Rating:** 5
**Confidence:** 4

**Summary:**

The authors propose a frequency-aware MAE design for multimodal pretraining on time-series biosignals in the frequency domain. They use a frequency-aware transformer encoder and a frequency-maintain pretraining strategy with masked autoencoding in the latent space. Specifically, the proposed encoder mainly consists of a frequency-domain feature extractor after DFT transformation and an attention-based dynamic fusion mechanism. The frequency-maintain pretraining strategy mainly addresses that masked autoencoding happens in the latent space and channel-independent feature encoding. Experimental results on uni-modal and multi-modal biosignals are provided to demonstrate the advantage of the proposed approach.

**Strengths:**

1. The authors tackled an interesting and practical problem in the domain of multi-modal pretraining of time-series sensing signals, with resiliency and robustness considerations.

2. Extensive evaluation results on both uni-modal and multi-modal scenarios are reported.

3. The presentation and writing of the paper is of good quality. It is generally easy to follow and understand the paper.

4. The robustness test presented in the experiment helps justify the performance of the approach.

**Weaknesses:**

1. The authors claimed the frequency-domain analysis and feature extraction for biosignals (being time series) as one of the main contributions in this paper, which actually have been well studied in the sensing community [1, 2]. Besides, the Transformer encoder, the masked autoencoding paradigm, and the attention-based fusion mechanism are original and novel contributions made in this paper. The channel-independent design, as mentioned in the paper, has been studied in several previous works. For this reason, I feel the novelty of the proposed approach is limited.

2. The approach of this paper seems to be generally applicable to various time-series sensing modalities, and there is no special design unique to the biosignals. I am not sure why you position the paper as one, especially for multi-modal pretraining on biosignals.

3. There lack of comparisons to existing multi-modal contrastive frameworks in the literature [3, 4], so it would be hard to understand its performance in the multi-modal collaboration scenario.

4. There is a comparison between channel-independent learning and conventional approaches that simultaneously consume all channels. In my personal opinion, channel-independent learning might be less efficient in computation (reduced parallelizability) and might not convey the global information (related to all channels) very well, which could be hard to model by simply concatenating extracted channel features.

[1] Yao, Shuochao, et al. "Stfnets: Learning sensing signals from the time-frequency perspective with short-time Fourier neural networks." The World Wide Web Conference. 2019.

[2] Li, Shuheng, et al. "Units: Short-time Fourier inspired neural networks for sensory time series classification." Proceedings of the 19th ACM Conference on Embedded Networked Sensor Systems. 2021.

[3] Tian, Yonglong, et al. "Contrastive multiview coding." Computer Vision–ECCV 2020: 16th European Conference, Glasgow, UK, August 23–28, 2020, Proceedings, Part XI 16. Springer International Publishing, 2020.

[4] Poklukar, Petra, et al. "Geometric multimodal contrastive representation learning." International Conference on Machine Learning. PMLR, 2022

**Questions:**

1. How many labels do you use during the fine-tuning stage? What finetuning strategy do you use? Do you only finetune the appended classification layers or update the whole encoder parameters?

2. What is the main design that distinguishes your approach from existing frequency-domain encoders for time-series signals? There is no discussion or experiment on this comparison in the paper.

3. Could you provide an ablation study on turning on/off the channel independent learning?

---

> ### Author Response · Authors · 2023-11-17
> **Response to Reviewer BHTD**
>
> Dear reviewer BHTD, we appreciate your time and thoughtful feedback. Especially, thank you for evaluating our work as “interesting and practical”, having “extensive evaluation”, and recognizing our “robustness tests”. We greatly appreciate your positive comments!
>
> ---
> ### **Novelty of our approach, and why we emphasize pretraining on biosignals**
>
> Thank you very much for pointing us to works in the sensing community! We will make sure to include and discuss them in the revised text. Also, thank you for recognizing the novelty of our transformer encoder, MAE paradigm, and attention-based fusion mechanism.
>
> Although the channel-independence technique **itself** isn’t novel, we would like to emphasize that our work aims to present a **multimodal pretraining strategy** that could (i) effectively combine information across different modalities including Electroencephalogram (EEG), Electrooculography (EOG), Electromyography (EMG), and Respiration rates (Resp) at the same time during pretraining, and (ii) could be flexibly applied on many other (unseen) modalities (e.g. Electromechanics). To the best of our knowledge, we are the first work that demonstrates that using information from different physiological modalities could improve the performance on downstream tasks.
>
> We agree that the proposed method is generally applicable to various time-series sensing modalities, and consider it as an advantage of our work. However, we think the challenges, and the enabled applications of our approach are unique in the biosignal field:
>
> 1. Firstly, different physiological signals are recorded using different types of sensors placed at various locations on the body. The diversity in sensor technology and placement differentiates them as unique modalities. The signals also differ significantly in terms of their signal characteristics, sampling rate, and effective frequency bands.
>
> 2. Secondly, each signal type is affected differently by various types of noise and artifacts, causing large distributional shifts within each modality. In practice, the available combination of those signals also varies, causing large distributional shifts across modalities. The challenges in mapping, aligning, and analyzing data under such distributional shifts are central to research in physiological signal processing.
>
> For the novelty of our approach, aside from the motivation that we clarified above, we would like to emphasize that our approach is technically distinguished from other works from two perspectives:
>
> 1. The most innovative aspect of bioFAME is its use of a fixed-size frequency filter. Inspired by FNO, our approach is specifically designed for biosignals and is significantly different from methods used in vision. The approaches in other works rely on frequency kernel of size $C^{N \times D}$, where N is the number of patches and D is the dimensionality of each patch. Thus, it cannot be directly transferred on data of different sizes, while our method, which learns queryable frequency filters of size $C^{K \times D}$, is adept at handling biosignals of varying lengths (e.g. 178 for Epilepsy, 5120 for FD-B) without performance degradation.
>
> 2. Our model is also different from Fourier-based approaches that were previously used in physiological signals and the sensing community. Our approach **does not** rely on explicit frequency-space encoder or frequency-space decoder, and does not use reconstruction loss in the frequency space. We found our method more flexible, parameter-efficient, and also easier to implement than other approaches.
>
> We hope the novelty of the proposed approach isn’t shadowed due to the channel-independence technique we used to improve transferability. Such techniques are carefully selected and combined together to ensure that the final architecture is transferable and flexible!

---

> ### Author Response · Authors · 2023-11-17
>
> ### **Other multi-modal contrastive frameworks as baselines**
>
> We thank the reviewer for their suggestion. To the best of our knowledge, no existing multimodal contrastive framework demonstrated effective performance on multimodal physiological signals. Existing work [1] even suggests that training on multiple different physiological signals gives worse performance, due to the large variation across different signals. We believe that (i) the lack of such baselines **highlights the contribution of our work**, as we are the first work that could effectively use many modalities (EEG, EMG, EOG, Resp) for pretraining; (ii) it might be out of our scope to implement such baselines.
>
> [1] Zhang, Xiang, et al. "Self-supervised contrastive pre-training for time series via time-frequency consistency." Advances in Neural Information Processing Systems 35 (2022): 3988-4003.
>
> ---
> ### **The pros and cons of channel-independent learning**
>
> We are grateful for your insightful comments on the computational efficiency and global information representation in channel-independent learning. We agree that the usage of channel independence is a balance between transferability and capacity. To address the capacity concern, bioFAME incorporates a lightweight encoder (Encoder 2) in its architecture. We still selected channel independence to ensure transferability, but the lightweight encoder during pretrained helped the information across different channels and modalities mix and integrate. We note that, without channel independence, many of our experiments cannot be conducted due to the shifts and mismatch of the total amount of channels.
>
> ---
> ### **Details of experimental settings during fine-tuning stage**
>
> Thank you for your question about the detailed experimental settings, we will make sure to revise the text to clarify such experimental details! For the amount of labels used during fine-tuning stage, we presented the information in Appendix Table 7. We conducted full-scale fine-tuning instead of linear probing, and applied such fine-tuning pipeline to all baselines, because we believe it is a more common practice for transformer backbones [1], and is also more effective for contrastive learning methods [2].
>
> [1] He, Kaiming, et al. "Masked autoencoders are scalable vision learners." Proceedings of the IEEE/CVF conference on computer vision and pattern recognition. 2022.
>
> [2] Chen, Ting, et al. "A simple framework for contrastive learning of visual representations." International conference on machine learning. PMLR, 2020.

---

> ### Author Response · Authors · 2023-11-21
> **Follow-up to the Rebuttal**
>
> Dear reviewer,
>
> Thank you once again for your insightful feedback. We have carefully revised our manuscript in line with your suggestions and believe we have comprehensively addressed your concerns. Could you kindly confirm whether the revisions align with your expectations? We are available to engage in further discussion or answer any additional questions you may have.
>
> Thanks!

---

> > ### Comment · Reviewer_BHTD · 2023-11-23
> > **Reply to the Authors' Responses**
> >
> > Thanks to the authors for your responses. Upon carefully reading your explanations and justifications, as well as comments from other reviewers, my concerns related to its novelty remain. Therefore, I would like to keep my original rating.

---

### Official Review · Reviewer_YFSy · 2023-11-01

**Soundness:** 3 good
**Presentation:** 2 fair
**Contribution:** 3 good
**Rating:** 8
**Confidence:** 3

**Summary:**

The authors present a Transformer-inspired architecture to learn representations of biosignal time-series in unimodal and multimodal settings with a masked autoencoding pretraining task. The proposed Frequency-aware block performs token mixing in the frequency domain while the masked autoencoding task in the learned latent space is meant to preserve the structure of the learned frequency-based representation during pretraining. The model is pretrained on the SleepEDF dataset (on EEG and/or EMG and EOG channels) and then finetuned on different downstream tasks that contain different biosignal modalities (e.g. EMG, EOG, electromechanical measurements). Evaluation in different settings, e.g. with unimodal pretraining, multimodal pretraining, and with modality substitution or dropout, shows the proposed approach outperforms existing baselines and is robust to changing or missing data.

**Strengths:**

Originality: The combination of a new frequency-based Transformer-like block along with a masked autoencoding pretraining task that enables multimodality processing appears novel.

Quality: The manuscript is overall of good quality. The proposed methodology is well-motivated and evaluated in relevant settings.

Clarity: The text and results are mostly clearly presented, however some core concepts would benefit from a clearer presentation (see Weaknesses).

Significance: The presented results which suggest that pretraining on EEG allows a substantial improvement on downstream tasks on different modalities is impressive (in some cases with significantly higher results than reported baselines). If the reported results hold in different multimodal/transfer settings (see Weaknesses) the proposed approach might be very useful on various multimodal tasks.

**Weaknesses:**

- The evaluation seems a bit limited given the claims in the abstract. Since the models are only pretrained on sleep data, I wonder if the downstream performance might reflect the nature of sleep data (or of the specific SleepEDF dataset) rather than the transferability of the representations. Reporting results on more diversified pretraining datasets (e.g. pretraining on epilepsy data to later test on sleep data) would ensure the proposed approach generalizes to additional settings.

- Although the text is written clearly, some parts of the pipeline would benefit from additional explanations on the initial MLP (see Q1 below) and the architecture of the second encoder (Q3).

**Questions:**

1. What does the initial MLP described in Section 4.1 end up learning to do in practice (intuitively and/or actually)? My understanding is that it is mixing temporal information within each patch $s_i$, which would already scramble (some of) the temporal information in higher frequencies that could be relevant for the task and common across modalities. Second, are the MLPs shared between modalities? If the sampling rates are not the same then I assume different MLPs would be required unless patches capture different time lengths depending on the modalities. On this topic, what are $N$ and $P$ in practice for the different modalities/datasets?

2. What is the architecture of the “second encoder” as analyzed in Table 3? To confirm, is this the same thing as the “lightweight” encoder of Section 4.2, first paragraph? Its exact role is unclear to me.

3. Have the authors tried a linear probing evaluation instead of full finetuning as in Table 1? This would provide a clearer evaluation of whether the model truly learned a “generic” representation of biosignal time series during pretraining.

4. How many parameters does the final encoder model(s) contain?

5. Did the authors evaluate the impact of using normalization (e.g. layer norm) and/or positional/temporal encoding as is commonly done with Transformers, or does the frequency-aware module not require such components?

---

> ### Author Response · Authors · 2023-11-17
> **Response to Reviewer YFSy**
>
> Dear reviewer YFSy, we appreciate your time and insightful feedback. Especially, thank you for evaluating our work as “novel”, “well-motivated and evaluated”, and has “substantial improvements that are impressive”. We greatly appreciate your positive comments!
>
> ---
> ### **Additional experiments to further support the claims**
>
> We believe your comment “models are only pretrained on sleep data, … might reflect the nature of sleep data” is very insightful and spot on. We do believe that the performance of our model would be greatly affected by its pretraining dataset, similar as foundation models in many fields [1, 2, 3].
>
> However, we believe that this would not dim the effectiveness of our approach. To validate that our pretraining strategy outperforms other strategies under diverse pretraining datasets, we conducted **two additional experiments**: we pretrain bioFAME, and a competitive baseline PatchTST on (i) FDA dataset; (ii) TUH EEG dataset, and transfer the learned representation on new datasets.
>
> The below table shows the collected results, where bioFAME (S) represents the SleepEDF pretrained model for benchmarking; bioFAME (FDA) represents the FDA pretrained model; and bioFAME (TUH) represents the TUH EEG pretrained model.
>
> | Methods | FD-B   | Epilepsy | ExpEMG | Avg |
> | -------- | ------- | -------- | ------- | ------- |
> | PatchTST (S) | 67.03 | 95.01 | 92.68 | 84.91 |
> | bioFAME (S) | 76.58 | 95.51 | 98.05 | 90.05 |
> | $\Delta$ | $\uparrow$ 9.55 | $\uparrow$ 0.50 | $\uparrow$ 5.37 | $\uparrow$ 5.14 |
>
> | Methods | FD-B   | Epilepsy | ExpEMG | Avg |
> | -------- | ------- | -------- | ------- | ------- |
> | PatchTST (FDA) | 88.42 | 84.95 | 89.45 | 87.61 |
> | bioFAME (FDA) | 90.46 | 92.94 | 95.55 | 92.98 |
> | $\Delta$ | $\uparrow$ 2.04 | $\uparrow$ 7.99 | $\uparrow$ 6.10 | $\uparrow$ 5.37 |
>
> | Methods | FD-B   | Epilepsy | ExpEMG | Avg |
> | -------- | ------- | -------- | ------- | ------- |
> | PatchTST (TUH) | 50.08 | 82.98 | 78.65 | 70.57 |
> | bioFAME (TUH) | 77.91 | 90.02 | 94.58 | 87.50 |
> | $\Delta$ | $\uparrow$ 27.83 | $\uparrow$ 7.04 | $\uparrow$ 15.93 | $\uparrow$ 16.93 |
>
> There are two major observations:
> (i) bioFAME, as a robust pretraining strategy, outperforms the PatchTST baseline given the same pretraining data in diverse situations.
> (ii) The representation quality of bioFAME highly correlates with the quality of pretraining dataset, and our choice of multimodal sleep dataset gives high-quality representation. Interestingly, pretraining on FDA dataset gives large performance gain on FDB downstream task; while pretraining on the TUH EEG corpus in general gives worse performance, probably due to limited training time (see details below).
>
> Below are the more specific training & dataset details:
>
> 1. FDA dataset is another corpus of single-channel Electromechanics dataset. We divide the signal into chunks of length 5120, creating a training dataset of sample size 8,000, and pretrain the network on it for 200 epochs with learning rate 1e-3 using an Adam optimizer for both models.
>
> 2. TUH EEG is a large-scale abnormal EEG corpus. We preprocess the signal following a public GitHub repo https://github.com/AITRICS/EEG_real_time_seizure_detection, and then chunk the signal into samples of length 4000, creating a 20-channel unimodal (EEG) dataset of sample size 102,903. Due to the limited amount of time, we pretrain the network for 10 epochs with learning rate 1e-3 using an Adam optimizer for both models.
>
> We identify that large amount of high-quality dataset for pretraining is one of the major challenges of the biosignal field, and hypothesize that sleep data, with its diverse modalities and long period of time available, is an appropriate testbed for our proposed method. We hope our approach, by combining information across modalities for pretraining, can further advocate and encourage the production of such datasets, and benefit the physiological research field.
>
> [1] Touvron, Hugo, et al. "Llama 2: Open foundation and fine-tuned chat models." arXiv preprint arXiv:2307.09288 (2023).
>
> [2] Kirillov, Alexander, et al. "Segment anything." arXiv preprint arXiv:2304.02643 (2023).
>
> [3] Radford, Alec, et al. "Learning transferable visual models from natural language supervision." International conference on machine learning. PMLR, 2021.

---

> > ### Author Response · Authors · 2023-11-17
> >
> > ### **Additional clarifications and explanations of our training pipeline**
> >
> > Thank you for pointing out parts that need further improvement in clarity! We answer your questions as below, and will make sure to revise the text to ensure the clarity and readability.
> >
> > - Role of the initial MLP.
> >
> > We believe that the initial MLP, as a tokenization layer, does mix temporal information in the high-frequency. We also do share the MLP between different modalities for now. While we believe it might be preferable to train individual MLP for different modalities, however, that would limit our capacity to transfer the proposed architecture to other (especially, new) modalities. To mitigate the drawbacks of that, we selected a small number of patch size (P=20) so that the MLP could generalize in different situations, and the additional ablation shows that increasing the latent dimensionality (dim=256) would also increase the performance. We hypothesize that the overparameterization in the tokenization stage is one of the keys to our approach.
> >
> > - Role of the second encoder.
> >
> > The reviewer is correct, the “lightweight” encoder is the same thing as the second encoder, we will rephrase it in the main text to improve clarity. The exact role of it is to encourage information mixing under the channel-independent design (of the first encoder) in the pretraining stage.
> >
> > ---
> > ### **Linear probing v.s. Full-scale finetuning**
> >
> > We conducted full-scale fine-tuning instead of linear probing, because we believe it is a more common practice for transformer backbones, especially when trained under the reconstruction loss. This observation is provided in [1]. To make the comparison fair, we also conducted full-scale fine-tuning for other baseline models, as it is shown that it is more effective than linear probing for contrastive learning methods as well [2]. Given the setting, we think it still shows the learned representation is beneficial to downstream tasks.
> >
> > [1] He, Kaiming, et al. "Masked autoencoders are scalable vision learners." Proceedings of the IEEE/CVF conference on computer vision and pattern recognition. 2022.
> >
> > [2] Chen, Ting, et al. "A simple framework for contrastive learning of visual representations." International conference on machine learning. PMLR, 2020.
> >
> > ---
> > ### **Parameter comparison**
> >
> > To understand the size of the model, we compare the total number of parameters (Params) and FLOPs of our approach and baselines.
> >
> > | Methods | TS2vec  | TFC | TS-TCC | PatchTST | **bioFAME** (Ours) |
> > | -------- | ------- | -------- | ------- | ------- | ------- |
> > | Params | 632K | 1.18M | 140K | 612K | 243K |
> > | FLOPs | 0.69B | 1.38B | 1.95B | 35.0B | 9.42B |
> >
> > We can see that, bioFAME is very parameter-efficient due to its fix-size frequency filter design. With the same depth (4), heads (8), and dimensionality (64), bioFAME contains only ~40% parameters of the transformer baseline PatchTST. The parameter size of bioFAME also stands competitive than many CNN-based architectures, with only TS-TCC smaller than it.
> >
> > To gain a better understanding of the throughput of the model, we used the fvcore package to compute the FLOPs of the models over a batch of SleepEDF data (batch size = 64). We found that the FLOPs of bioFAME is significantly lower than the transformer baseline PatchTST (<30%); yet greater than CNN-based architectures.
> >
> > ---
> > ### **The impact of normalization and positional embedding**
> >
> > Thank you for asking for the details of the implementation, we will make sure to include these details during the revision! We did use layer normalization for both transformers, and use the positional embedding in the second encoder during the reconstruction phase. We did not use positional embedding in the frequency-aware encoder because we found them not improving the performance of the transfer experiments.

---

> > > ### Comment · Reviewer_YFSy · 2023-11-21
> > >
> > > Thank you to the authors for their answers. Training pipeline and model architecture details are clearer now and the additional results on different pretraining datasets answers my main question. I am improving my score to 8.

---

> > > > ### Author Response · Authors · 2023-11-21
> > > >
> > > > Thank you!

---

### Official Review · Reviewer_GTFm · 2023-11-06

**Soundness:** 3 good
**Presentation:** 3 good
**Contribution:** 2 fair
**Rating:** 3
**Confidence:** 4

**Summary:**

This paper investigates the pre-training on bio-signals (1D time series neurological signals) based on Masked Modeling strategies like MAE. To effectively pre-training with the bio-signals with potential distributional shifts, the authors design a frequency-aware masked auto-encoder, dubbed bioFAME, to learn bio-signal representations in the Fourier domain. Specifically, a frequency-aware transformer is designed, which leverages a fixed-size Fourier-based operator for token mixing. Meanwhile, the authors propose a frequency-maintain pre-training strategy that performs masking and reconstruction in the latent space to sustain the frequency components. Conducting pre-training and fine-tuning experiments, results on five bio-signal datasets show performance gains of the uni-modality and multi-modality versions of bioFAME.

**Strengths:**

(**S1**) This paper proposes a frequency-aware Transformer architecture and a latent-space masked auto-encoder pre-training strategy for bio-signals. Analysis and evaluation of the pre-training strategy are extensive. Experiment results demonstrate the effectiveness of both the proposed methods with performance gains and transferring abilities.

(**S2**) The overall writing is easy to follow, and the presentation of texts, tables, and figures is well-arranged and informative.

**Weaknesses:**

(**W1**) Lack of novelty and discussion with existing methods. The proposed method borrows the idea of MAE with the frequency modeling and reconstruction of neurological signals. The proposed frequency-aware transformer seems to borrow a similar design from FNO [1] and Ge$^2$AE relevant works [2, 3, 4] in computer vision. More importantly, the idea of this paper is quite similar to neuro2vec [5] proposed in 2022, which designs a masked modeling pre-training framework for neurological signals in both the spatiotemporal and Fourier domains and conducts experiments on open-source neurological datasets. The authors should discuss the relationship between these existing works and the proposed bioFAME, and point out the necessity of designing the frequency-aware modeling framework for bio-signal representations.

(**W2**) The proposed frequency-aware Transformer encoder lacks analysis. Since the architecture also brings significant performance gains compared to previous networks, the details of network design should be discussed soundly, e.g., the layer number, the embedding dimension, and the utility of the multi-head frequency filter layer.

(**W3**) Concerns about performance gains and experimental settings. Firstly, Compared to prior arts, the improvement of bioFAME is marginal. Also, since not all arts utilize multimodal training, it is only fair to compare bioFAME unimodal with other approaches. The experimental setting of unimodal vs multimodal is very confusing. The authors state that ExpEMG is a dataset of single-channel EMG recordings, how is the multimodal approach applied in this case? This same problem also applies to other datasets.
Secondly, bio-signal datasets are usually of small size, which leads to the problem of high variance over different random seeds. How did the authors manage to solve this? The authors claim to follow the setup of Zhang et al. [6], yet they [6] reported mean and variance. Why did the authors fail to do so?
Thirdly, since there are contrastive learning baselines (e.g., TS-TCC), do the authors conduct linear evaluations of the learned representations (e.g., linear probing)? Meanwhile, it is better to provide more empirical analysis of the learned embedding, e.g., embedding visualization by tSNE, and reconstruction result visualizations.

(**W4**) The connection between the motivation and approach is vague. The authors claim substantial distributional shifts between the pretraining and inference datasets. How can FREQUENCY-AWARE MASKED AUTOENCODERS solve this? What is expected to be learned from FREQUENCY-AWARE MASKED AUTOENCODERS, and how is this linked to the architecture design of the model?

(**W5**) Details of pre-training and fine-tuning settings on each dataset of bioFAME are vague. Hyper-parameter settings and sensitivity analysis are not available in the main text and appendix.

### Reference
[1] Li et al. Fourier neural operator for parametric partial differential equations. In ICLR, 2021.

[2] Liu et al. The Devil is in the Frequency: Geminated Gestalt Autoencoder for Self-Supervised Visual Pre-Training. In AAAI, 2023.

[3] Li et al. Architecture-Agnostic Masked Image Modeling - From ViT back to CNN. In ICML, 2023.

[4] Xie et al. Masked Frequency Modeling for Self-Supervised Visual Pre-Training. In ICLR, 2023.

[5] Wu et al. neuro2vec: Masked Fourier Spectrum Prediction for Neurophysiological Representation Learning. In arXiv, 2022.

[6] Zhang et al. Self-supervised contrastive pre-training for time series via time-frequency consistency. In NeurIPS, 2022.

**Questions:**

(**Q1**) Why did the authors call bioFAME multimodel pre-training? Training on multiple channels on the same modality should not be called multimodal.

(**Q2**) Did the authors provide details of pre-training and fine-tuning settings on each dataset of bioFAME? Hyper-parameter settings and sensitivity analysis are not available in the main text and appendix.

================== Post-rebuttal Feedback ==================

Thanks for the detailed rebuttal feedback. However, my main concerns about novelty were not well solved, and I decided to maintain my rating. I apologize again for my late review and reply!

---

> ### Author Response · Authors · 2023-11-17
> **Response to Reviewer GTFm**
>
> Dear reviewer GTFm, we appreciate your time and valuable comments. Especially, thank you for evaluating our experimental analysis as “extensive”, and recognizing the effectiveness of our proposed method.
>
> In response to the major criticism about multimodal experimental settings and the confusion regarding “how is the multimodal approach applied” to “single-channel EMG recordings”, we must clarify that the goal of the paper is to present a **multimodal pretraining strategy** that could (i) effectively combine information across different modalities including Electroencephalogram (EEG), Electrooculography (EOG), Electromyography (EMG), and Respiration rates (Resp) at the same time during pretraining, and (ii) could be flexibly applied on many other (unseen) modalities (e.g. Electromechanics). This is fundamentally different from works like neuro2vec [1], which represents a robust strategy that pretraining and testing on the same physiological modality, and does not evaluate how much adding additional information during pretraining could further help with building robust representation. We hope our approach can fundamentally address the challenge and unify learning on many different physiological signals, which is a problem that has not been systematically tackled before in this community.
>
> [1] Wu, Di, et al. "neuro2vec: Masked Fourier spectrum prediction for neurophysiological representation learning." arXiv preprint arXiv:2204.12440 (2022).
>
> ---
> ### **W1 - Novelty of our approach and additional discussion with existing methods**
>
> For the novelty of our approach, aside from the motivation that we clarified above, we would like to emphasize that our approach is technically distinguished from other works from two perspectives:
>
> 1. The most innovative aspect of bioFAME is its use of a fixed-size frequency filter. Inspired by FNO [1], our approach is specifically designed for biosignals and is significantly different from methods used in vision [2, 3, 4]. The approaches in vision rely on frequency kernel of size $C^{N \times D}$, where N is the number of patches and D is the dimensionality of each patch. Thus, it cannot be directly transferred on data of different sizes, while our method, which learns queryable frequency filters of size $C^{K \times D}$, is adept at handling biosignals of varying lengths (e.g. 178 for Epilepsy, 5120 for FD-B) without performance degradation.
>
> 2. Our model is also different from Fourier-based approaches that were previously used in physiological signals [5]. Our approach **does not** rely on explicit frequency-space encoder or frequency-space decoder, and does not use reconstruction loss in the frequency space. We found our method more flexible, parameter-efficient, and also easier to implement than other approaches.
>
> We thank the reviewer for the comment one more time. We will revise the manuscript to extend the discussion of our approach and its differences with existing methods. We will also make sure to cite and discuss papers [3, 5], as [1, 2, 4] are already discussed in the related works and method preliminaries.
>
> [1] Li et al. Fourier neural operator for parametric partial differential equations. In ICLR, 2021.
>
> [2] Liu et al. The Devil is in the Frequency: Geminated Gestalt Autoencoder for Self-Supervised Visual Pre-Training. In AAAI, 2023.
>
> [3] Li et al. Architecture-Agnostic Masked Image Modeling - From ViT back to CNN. In ICML, 2023.
>
> [4] Xie et al. Masked Frequency Modeling for Self-Supervised Visual Pre-Training. In ICLR, 2023.
>
> [5] Wu et al. neuro2vec: Masked Fourier Spectrum Prediction for Neurophysiological Representation Learning. In arXiv, 2022.

---

> > ### Author Response · Authors · 2023-11-17
> >
> > ### **W2, W5 - Additional analysis and ablations**
> >
> > To further understand the details of network design, we first computed the parameters and FLOPs of bioFAME to validate that the comparison is fair.
> >
> > | Methods | TS2vec  | TFC | TS-TCC | PatchTST | **bioFAME** (Ours) |
> > | -------- | ------- | -------- | ------- | ------- | ------- |
> > | Params | 632K | 1.18M | 140K | 612K | 243K |
> > | FLOPs | 0.69B | 1.38B | 1.95B | 35.0B | 9.42B |
> >
> > We can see that, bioFAME is very parameter-efficient due to its fix-size frequency filter design. With the same depth (4), heads (8), and dimensionality (64), bioFAME contains only ~40% parameters of the transformer baseline PatchTST. The parameter size of bioFAME also stands competitive than many CNN-based architectures, with only TS-TCC smaller than it.
> >
> > To gain a better understanding of the throughput of the model, we used the fvcore package to compute the FLOPs of the models over a batch of SleepEDF data (batch size = 64). We found that the FLOPs of bioFAME is significantly lower than the transformer baseline PatchTST (<30%); yet greater than CNN-based architectures.
> >
> > To understand the sensitivity of hyper-parameters, we conducted additional ablations to study if our model could benefit from increased dimensionality and network depth.
> >
> > | Latent dim | 32 | 64 | 128  | 256 |
> > | -------- | ------- | -------- | ------- | ------- |
> > | EMG | 91.1 | 98.05 | 96.48  | 97.78  |
> > | FDB | 76.74  | 76.58 | 78.14  | 80.87  |
> > | Avg. | 83.92 | 87.32 | 87.31 | 89.33 |
> >
> > | Network Depth | 3 | 4 | 5  | 6 |
> > | -------- | ------- | -------- | ------- | ------- |
> > | EMG | 77.54 | 76.58 | 76.79 | 78.99 |
> > | FDB | 97.78 | 98.05 | 95.55 | 92.59 |
> > | Avg. | 87.66 | 87.32 | 86.17 | 85.79 |
> >
> > We observed that increasing the latent dimensionality could further improve the performance of our approach; while increasing the network depth gives no performance gains.
> >
> > We would like to thank the reviewer for pushing us to consider these additional ablations. We hope the additional ablation results, together with the ablation results in the original paper (Specifically, the use of the multi-head frequency filter layer in Table 2, and the amount of filters in Appendix Figure 4B), are sufficient for demonstrating the strength and stability of the approach.
> >
> > ---
> > ### **W3 - Clarification about model variance and performance gains**
> >
> > We would like to clarify that we have included model standard deviation towards different random seeds in Appendix Table 5, which follows the practice of Zhang et al. [6]. We only included this table in Appendix due to space constraints, and pointed to it in the caption of Table 1 in the main text.
> >
> > Additionally, we would like to point out that the improvement of bioFAME is beyond marginal even under the unimodal setting, that is what we compared and stated in the “Pretraining on Unimodality” section on Page 7 in the main text. A re-organized version of Table 1 is as below, we will also revise Table 1 in the main text to make it clearer.
> >
> > |  | Epilepsy | SleepEOG | ExpEMG  | FD-B |
> > | -------- | ------- | -------- | ------- | ------- |
> > | Previous SOTA | 95.07 | 69.65 | 92.68  | 69.38  |
> > | bioFAME (unimodal) | 95.51 | 70.03  | 98.05  | 76.58  |
> > | $\Delta$ | $\uparrow$ 0.44 | $\uparrow$ 0.38 | $\uparrow$ 5.37 | $\uparrow$ 7.20 |

---

> > > ### Author Response · Authors · 2023-11-17
> > >
> > > ### **Q1, W4 - Further clarification about motivation**
> > >
> > > Thank you for expressing your concern over our motivation. We believe that EEG, EOG, EMG, and Respiration belong to different modalities, due to the following facts:
> > >
> > > (i) Firstly, these biosignals are recorded using different types of sensors placed at various locations on the body. The diversity in sensor technology and placement differentiates them as unique modalities. The signals also differ significantly in terms of their signal characteristics, sampling rate, and effective frequency bands.
> > >
> > > (ii) Secondly, each signal type is affected differently by various types of noise and artifacts, causing large distributional shifts within each modality. In practice, the available combination of those signals also varies, causing large distributional shifts across modalities. The challenges in mapping, aligning, and analyzing data under such distributional shifts are central to research in physiological signal processing.
> > >
> > > Our work highlights the benefit of pretraining on many modalities to performance improvement on various downstream tasks with seen or unseen modalities. We will further edit the motivation section to make our point clearer, and hope that helps!
> > >
> > > ---
> > > ### **Q2, W5 - Additional details about experimental settings**
> > >
> > > We would like to express our gratitude for the kind feedback from the reviewer. We will add an additional section in the Appendix to be very specific about the experimental settings.
> > >
> > > We thank the reviewer for their detailed comments, and hope our clarification helps! Please let us know if there is other unclear parts and we will try our best to answer.

---

> ### Author Response · Authors · 2023-11-21
> **Follow-Up to the Rebuttal**
>
> Dear reviewer,
>
> Thank you once again for your insightful feedback. We have carefully revised our manuscript in line with your suggestions and believe we have comprehensively addressed your concerns. Could you kindly confirm whether the revisions align with your expectations? We are available to engage in further discussion or answer any additional questions you may have.
>
> Thanks!

---

> ### Comment · Reviewer_GTFm · 2023-11-23
> **Response to Authors' Rebuttal**
>
> Thanks for the detailed and comprehensive responses, and sorry for the late reply at the end of the rebuttal period! I appreciate the technical contributions and well-organized writing of this work. However, the authors' responses have not addressed all my concerns, and I list them as follows:
>
> * (W1, W4, Q1) The novelty is my main concern. Firstly, I knew that these works (e.g., works [1-4] mentioned in the authors' responses) I mentioned were proposed in different fields, and this work aims to tackle the pre-training problem of bio-signals in multi-modalities. However, as a new method for the AI community, it is necessary to provide strong motivations and reasons if the new method directly utilizes existing techniques to tackle special problems in downstream applications. Otherwise, Otherwise, I can arrange and combine some related techniques to make another new paper. I suggest the authors clarify these points in the main text (better to use empirical analysis and comparison). Secondly, the relationship and difference between bioFAME and neuro2vec [5] mentioned in the response are not convincing to me. And, I suggest the authors add this discussion in the related works (or some sections in the appendix).
>
> * (W2, W5) These issues were well addressed and added to the revised version. A minor issue is that the FLOPs of bioFAME are relatively high, which will be a limitation of the proposed network architecture.
>
> * (W3, W5, Q2) These issues were almost addressed. A minor issue is the performance gains of bioFAME are not significant on Epilepsy and SleepEOG. Is there any explanation? Since the performance gain of bioFAME comes from both the network architecture and the pre-training, it is better to clarify whether bioFAME can consistently achieve superior performances on different scenarios, or the limitations should be discussed and explained.
>
> Overall, I decided to keep my current rating based on the above issues.

---

### Author Response · Authors · 2023-11-17
**General response**

We would like to thank all reviewers for their encouraging and constructive feedback. Specifically, we want to thank **all reviewers** for recognizing the *effectiveness of our approach*, and we want to thank reviewers YFSy and hfvK for recognizing our work as *‘novel’*, and reviewer BHTD for recognizing our work as *‘interesting’*.

The insightful comments and feedback from reviewers have **improved our submission substantially**. We were able to conduct additional experiments to address reviewers’ comments and concerns, further improve the clarity of our manuscript, and justify the contribution and innovation of bioFAME. While we will be responding to each reviewer in detail, we outline the major additions to the current manuscript as below:

----
### **Motivation and distinction with existing works**

- We would like to emphasize that the contribution of our work is beyond the design of a new frequency-space modeling approach (concerns of reviewer GTFm and BHTD). Instead, our approach is the **first work** that presents a **multimodal pretraining strategy** that could (i) effectively use and combine information across many different modalities including Electroencephalogram (EEG), Electrooculography (EOG), Electromyography (EMG), and Respiration rates (Resp) **at the same time** during pretraining, and (ii) could be flexibly applied on many other (even unseen) modalities (e.g. Electromechanics). This is fundamentally different from existing works, which either pretrain and test on the same physiological modality [1], or fail to show that adding additional modalities during pretraining could further improve the quality of the representation [2]. We hope our work, focusing on pretraining on multimodal biosignals, can pave the way and raise the community’s attention towards building unified representation for different physiological signals.

- We would like to express our great appreciation to the reviewers’ comments to help us realize that we could discuss our motivation and contribution clearer, and we should include additional comparison and discussion with existing works in the main text. Thus, we plan to revise the Related Work Section and cite/discuss the works pointed out by reviewers to improve the paper.

[1] Wu, Di, et al. "neuro2vec: Masked Fourier spectrum prediction for neurophysiological representation learning." arXiv preprint arXiv:2204.12440 (2022).

[2] Zhang, Xiang, et al. "Self-supervised contrastive pre-training for time series via time-frequency consistency." Advances in Neural Information Processing Systems 35 (2022): 3988-4003.

---
### **bioFAME is parameter-efficient, and stable across different hyperparameters**

- All reviewers asked about the parameter size of the proposed method, and some raised concerns about the fairness of comparison due to unknown parameter size. To answer this, we computed the parameter size and FLOPs of bioFAME as below:

| Methods | TS2vec  | TFC | TS-TCC | PatchTST | **bioFAME** (Ours) |
| -------- | ------- | -------- | ------- | ------- | ------- |
| Params | 632K | 1.18M | 140K | 612K | 243K |
| FLOPs | 0.69B | 1.38B | 1.95B | 35.0B | 9.42B |

We can see that, bioFAME is very parameter-efficient due to its fix-size frequency filter design. With the same depth (4), heads (8), and dimensionality (64), bioFAME contains only ~40% parameters of the transformer baseline PatchTST. The parameter size of bioFAME also stands competitive with many CNN-based architectures, with only TS-TCC smaller than it.

To gain a better understanding of the throughput of the model, we used the fvcore package to compute the FLOPs of the models over a batch of SleepEDF data (batch size = 64). We found that the FLOPs of bioFAME are significantly lower than the transformer baseline PatchTST (<30%); yet greater than CNN-based architectures.

- Reviewer GTFm and hfvK also asked about the models’ sensitivity towards different hyperparameters. To understand that, we add **two new ablation experiments** to investigate the effect of different depth and latent dimensionality.

| Latent dimension | 32 | 64 | 128  | 256 |
| -------- | ------- | -------- | ------- | ------- |
| EMG | 91.1 | 98.05 | 96.48  | 97.78  |
| FDB | 76.74  | 76.58 | 78.14  | 80.87  |
| Avg. | 83.92 | 87.32 | 87.31 | 89.33 |

| Network Depth | 3 | 4 | 5  | 6 |
| -------- | ------- | -------- | ------- | ------- |
| EMG | 77.54 | 76.58 | 76.79 | 78.99 |
| FDB | 97.78 | 98.05 | 95.55 | 92.59 |
| Avg. | 87.66 | 87.32 | 86.17 | 85.79 |

We observed that increasing the latent dimensionality could further improve the performance of our approach; while increasing the network depth gives no performance gains.

---

> ### Author Response · Authors · 2023-11-17
>
> ---
> ### **bioFAME is effective with diversified pretraining datasets**
>
> Reviewer YFSy pointed out that the contribution of our approach might be limited by the specific selection of pretraining dataset. To address this concern, we conducted **two new experiments** evaluating bioFAME’s performance under diversified pretraining datasets. Additional empirical results suggest that our approach, outperforming robust transformer baseline, is a powerful methodology for extracting biosignal representation regardless of the type of pretraining datasets.
>
> ---
> ### **More details about experimental setup**
>
> Additionally, we will go through the paper to further improve clarity in response to the reviewers’ comments and questions. Specifically, we will ensure all experimental details and setup are included to help the readers reproduce our results. We will also public our codebase soon for others to explore.

---

### Author Response · Authors · 2023-11-21
**Manuscript updated**

Dear reviewers,

We would like to express our gratitude for your encouraging and constructive feedback on our manuscript. In response to your suggestions, we have updated the manuscript, highlighting all changes in blue for ease of review. Below is a summary of the main revisions:

**Main text**
- We updated section 3 to further motivate our approach, and to extend the discussion on the differences from previous works.
- We revised the methods and results sections to improve the overall clarity.

**Appendix**
- We included the parameter comparison and the additional ablations in section A.1.
- We added the model training details and transfer experiments details in section B.2.
- The previously existing section on hyperparameters selection, section B.4, has been expanded to include more detailed information.

We would greatly appreciate your confirmation on whether the revisions have satisfactorily addressed your concerns. We are ready and willing to discuss any further questions you may have.

Thank you all!

---

### Meta-Review · Area_Chair_FGbK · 2023-12-06

**Metareview:**

The authors propose a frequency-aware MAE design for multimodal pretraining on time-series biosignals in the frequency domain. The proposed Frequency-aware block performs token mixing in the frequency domain while the masked autoencoding task in the learned latent space is meant to preserve the structure of the learned frequency-based representation during pretraining. Evaluation in different settings, e.g. with unimodal pretraining, multimodal pretraining, and with modality substitution or dropout, shows the proposed approach outperforms existing baselines and is robust to changing or missing data.


Strengths:
The use of frequency-aware MAE for multimodal biosignal pretraining is interesting. The evaluation on both uni-modal and multi-modal are useful to the community. The presentation and writing of the paper is of good quality.

Weaknesses:
There're some concerns raised by most reivewers on the novelty of the work. It probably be the first work on biosignals but technique wise the novelty is limited. Also it is unclear whether the proposed method is only effective to biosignal.

**Justification For Why Not Higher Score:**

The proposed frequency-aware MAE is interesting but the novelty is limited and it can be more convincing if the proposed methods can be validated on other multimodal datasets.

**Justification For Why Not Lower Score:**

N/A

---

### Decision · Program_Chairs · 2024-01-16

Reject